# FLAMES: MULTI-SCALE MAMBA WITH ADAPTIVE FOURIER FILTERS AND LAPLACE TRANSFORM FOR TIME SERIES FORECASTING

## ABSTRACT

Time series data usually exhibit intricate characteristics such as non-stationarity, noise, multi-scale periodicity, and transient dynamics, posing significant challenges to long-term time series forecasting (LTSF). While transformer-based models effectively capture long-range dependencies, their practical applications are hindered by high computational cost with quadratic complexity, noise sensitivity, and overfitting on small datasets. Moreover, time series present distinct patterns at different temporal resolutions, containing both fine-grained (micro) and coarse-grained (macro) information. To address these issues, we propose a novel framework, **Flames** (multi-**s**cale **F**ourier Filter **M**amba with **La**place), designed for efficient and robust LTSF. Specifically: (i) We introduce an adaptive Fourier filter with a selection module embedded into Mamba. At each scale, the neural operator uses Fourier analysis to refine feature representations, applies learnable thresholds for noise reduction, and captures inter-frequency interactions via global-local semantic filters through element multiplication. (ii) We incorporate the Laplace transform to capture transient dynamics. Extensive experiments on multiple benchmarks demonstrate that Flames consistently outperforms SOTA methods, achieving superior accuracy–efficiency trade-offs. Results highlight its strong robustness and scalability, particularly in noisy or transient settings. Code, data, and model checkpoints are in Supplementary Materials.

## 1 INTRODUCTION

Time series forecasting (TSF) aims to predict future temporal changes based on historical observations and is widely applied in finance, meteorology, healthcare, and transportation (Huang et al., 2024; Wu et al., 2024; Wang et al., 2024c; Long et al., 2024). Typically collected via IoT sensors and wearable devices, time series data often exhibit multi-scale periodicity, non-stationarity, and noise, posing challenges for long-term TSF (LTSF) (Wang et al., 2024b).

With the rise of deep learning, TSF has shifted from traditional statistical methods to neural network-based paradigms, dominated by four families: CNNs, RNNs, Transformers, and MLPs. CNNs excel at capturing local patterns but struggle with long-term dependencies (Zeng et al., 2024). RNNs, while well-suited for sequential modeling, suffer from parallel computing, low inference efficiency due to limited parallelism. Although LSTM and GRU improve long-term dependency modeling, they still face performance bottlenecks with ultra-long sequences. Transformers, empowered by a self-attention mechanism, offer a global receptive field and have become the mainstream in TSF, exemplified by models like PatchTST (Nie et al., 2023) and iTransformer (Liu et al., 2024b). However, they are prone to overfitting on small datasets, suffer from quadratic complexity (Liu et al., 2024b). Recent DLinear (Zeng et al., 2023) can surprisingly outperform complex Transformers in TSF, yet Linear struggles with noisy data and fails to capture long-term dependencies effectively.

State Space Model (SSM) has emerged as a promising alternative for modeling long-term dependencies. It typically employs ordinary differential equations to describe the states evolution over time, capturing long-range correlations with linear complexity and context-aware selectivity via hidden attention mechanisms (Cai et al., 2024b). This enables efficient inference over ultra-long sequences, balancing performance and computational cost, making SSMs well-suited for TSF. Mamba (Gu &

Dao, 2023) extends SSM with a selective attention-like mechanism to extract valuable information. Its linear state structure facilitates long-sequence modeling by propagating and forgetting information through input dependency selection. Mamba has shown success in visual expression (Zhu et al., 2024), language learning (Park et al., 2024), and image haze removal (Zheng & Wu, 2024). However, it still faces some challenges in TSF: (1) **Multiscale periodicity.** Time series often exhibit patterns at multiple time scales (e.g. daily and annual cycles in weather data). Effective modeling requires capturing both fine- and coarse-grained temporal patterns to reflect microscopic and macroscopic dynamics Shabani et al. (2022); Wang et al. (2024b). (2) **Data noise.** Time series are often noisy due to random variations, increasing the risk of overfitting, hampering meaningful signals extraction and stable performance Eldele et al. (2024); Yi et al. (2024a). (3) **Transient dynamics.** Time series may contain abrupt, short-lived changes driven by events or anomalies. Mamba's focus on point-wise dynamics restricts its capacity to model such transient behavior Zhang et al. (2025); Najda et al. (2025).

To tackle the aforementioned issues, we propose Flames, a novel framework tailored for LSTF, incorporating corresponding strategies to enhance Mamba's ability: (1) **Multi-scale feature extraction.** Coarser-grained sub-sequences are progressively constructed via average downsampling to learn multi-scale feature. (2) **Adaptive Fourier filter Mamba.** Mamba focuses on temporal dynamics but lacks frequency modeling and robustness to noise. To address this, we embed an adaptive Fourier filter (AFF) with a selection module into Mamba, which selectively attenuates high frequencies using learnable thresholds. AFF combines learnable global and local filters, applying Hadamard product-based adaptive filtering for full-frequency and high-frequency denoising. This enhances signal clarity and models both long- and short-term interactions, similar to circular convolution. (3) **Laplace transform.** To capture short-term fluctuations and event-driven changes, Laplace analysis is integrated with Mamba. The above technology enables Flames to capture multiscale periodicity, denosing, and transient behaviors, significantly enhancing the accuracy and efficiency of LSTF. With our detailed design, Flames consistently achieves SOTA performance across diverse benchmarks and maintains high efficiency. Our main contributions are as follows:

- We propose Flames, an efficient and universal time series forecasting framework.
- Flames excel in capturing multi-scale periodicity, transient dynamics, and noise reduction through average downsampling, the Laplace transform, and Fourier filtering. It contains one Mamba variant, employing an adaptive frequency filtering module (AFFM) with a selective block and global and local filters to cover all frequencies at each scale, dynamically learning thresholds to filter out noise. Then, the Laplace transform is to capture short-term fluctuations, further improving adaptability and generalization across diverse applications.
- Extensive experiments demonstrate that Flames achieves SOTA performance across various benchmarks, confirming its effectiveness and robustness in TSF.

## 2 RELATED WORKS

**State Space Models.** SSMs have shown strong potential in long-term sequence modeling. However, their high computational and memory demands often limit practical applications. To address this, several SSM variants have been proposed, such as S4 (Gu et al., 2021), S5 (Smith et al., 2022), H3 (Fu et al., 2022) and SSDNet (Lin et al., 2021). Recently, Mamba (Gu & Dao, 2023) advances SSM with an S4-based mechanism to filter irrelevant information and uses hardware-aware algorithms for parallel processing. It has proven success in computer vision (Tang et al., 2024), recommendation (Liu et al., 2024a), and graphics (Wang et al., 2024a). Besides, it achieves linear-time efficiency and outperforms Transformers in TSF (Cai et al., 2024b; Gu & Dao, 2023). For instance, MambaTS (Cai et al., 2024b) uses time-varying scanning to alleviate scanning-order sensitivity. TimeMachine (Ahamed & Cheng, 2024) integrates four Mamba blocks to jointly address channel mixing and independence. S-Mamba (Wang et al., 2024e) targets multivariate time series by modeling inter-sequence dependencies, while MambaMixer (Behrouz et al., 2024) introduces bidirectional blocks for both inter- and intra-sequence analysis. To further enhance its ability in TSF, our Flames could adaptively filter out noise, capture multi-scale interactions and transient dynamics.

**Frequency-Aware Time Series.** Frequency analysis, a fundamental technique in traditional signal processing (Baxes, 1994; Pitas, 2000), offers valuable insights by revealing frequency patterns. Some advances have explored frequency information to improve feature extraction (Yi et al., 2023).

Specifically, Autoformer (Wu et al., 2021) replaces self-attention with a Fourier auto-correlation mechanism. FEDformer (Zhou et al., 2022b) enhances attention using the discrete Fourier transform (DFT) to better capture long-term periodicity. Similarly, FiLM (Zhou et al., 2022a) applies Fourier analysis to filter out noise. FITS (Xu et al., 2023) utilizes rFFT and low-pass filters for compact representation, and TSLANet (Eldele et al., 2024) features adaptive spectral blocks to extract frequency features. However, many of them rely on manual feature engineering for cycle selection, often focusing only on dominant cycles and harmonics, which limits information diversity. Inspired by frequency filters in image processing and computer vision (Rao et al., 2021; Huang et al., 2023), we introduce learnable global and local filters in the Fourier domain to enhance semantic adaptability while reducing computational cost and parameter overhead.

## 3 METHOD

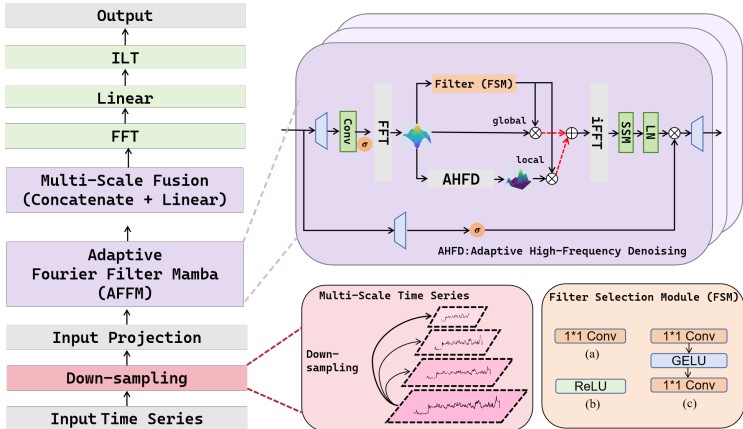

Figure 1: The structure of our proposed Flames.

As illustrated in Figure 1, our Flames includes three components: (1) obtaining multi-granularity time series on average; (2) AFFM encoder, using adaptive Fourier filter for multi-scale periodic pattern extraction and denoising; (3) Inverse Laplace transform to generate time-domain predictions.

### 3.1 MULTI-SCALE TIME SERIES.

We use a multi-scale framework for time series forecasting. Given an input series $x \in R^{T \times C}$, featuring a sequence length $T$ and $C$ variables. We progressively downsampled $x$ across M scales using convolution operations with a stride of $2^1$, generating a multi-scale set $X = \{x_0, ..., x_M\}$, where $\mathbf{x}_m \in R^{\lfloor \frac{T}{2^m} \rfloor \times C}$. The lowest-level series $x_0 = x$ is the input series, which contains the microscopic temporal variations, while the highest-level series $x_M$ is for the macroscopic variations. The downsampling process follows the recursive relationship:

$$x_m = Conv(x_{m-1}, stride = 2), m \in \{1, ..., M\}. \tag{1}$$

Then we project the multiscale series into deep features $X'$ by the embedding layer, which can be formalized as $X' = Embed(X) = \{x'_0, ..., x'_M\}$, where $\mathbf{x}'_m \in R^{\lfloor \frac{T}{2^m} \rfloor \times d_{model}}$, where $d_{model}$ denotes the dimensionality of the deep patterns. With these designs, we obtain the multiscale representation projection of the input series.

### 3.2 ADAPTIVE FOURIER FILTER MAMBA (AFFM).

To capture multi-scale periodicity and reduce noise, we propose an adaptive Fourier filter into Mamba to filter out noise, a limitation not addressed by existing S-Mamba and iTransformer. In Mamba, the step size $\Delta$ governs input dependency selection. However, all input signals are passed

through $\Delta$ without any filtering at each time step, leading to three key issues: (1) not all selected information is useful; (2) periodic patterns are difficult to capture; (3) $\Delta$ lacks the ability to distinguish noise. Based on the above reasons, we consider performing Fourier transform on $\Delta$ to better capture multi-scale information. Inspired by the convolution theorem (Eldele et al., 2024), we utilize the mathematical equivalence between time-domain convolution and frequency Hadamard product (i.e., element-wise multiplication). In this neural operator, AFFM identifies vital frequency components and filters out noise through four core steps:

(1) **Fast Fourier Transformations.** Transform the latent representation from the time domain to the frequency domain using 1D FFT operation: $X'_F = \mathcal{F}[X']$,

(2) **Adaptive High-Frequency Denoising (AHFD).** High-frequency components typically represent rapid fluctuations or deviations from the underlying trend, making them more random and less interpretable (Eldele et al., 2024). Removing such noise can help to identify multi-scale patterns, crucial for TSF. Besides, this operation can also simplify the model, speed up training, and reduce computational costs. Thus, we propose an adaptive high-frequency denoising filter with a learnable threshold. This filter suppresses irrelevant high-frequency noise while preserving essential information, yielding a more concise frequency representation, which is particularly valuable in non-stationary settings where the spectrum changes over time.

The trainable threshold $\theta_{high}$ is adaptively tuned to match the frequency properties, optimized through backpropagation to distinguish valuable information from high-frequency noise. Frequencies exceeding this threshold are removed, as formalized below:

$$X^{high}_{Filter} = X_F \odot (P > \theta_{high}) \tag{2}$$

where $\odot$ denotes element-wise multiplication (i.e., the Hadamard product), and $|F| \leq \theta_{high}$ means a binary mask where frequencies below $\theta_{high}$ are retained, while other frequencies are filtered out.

(3) **Learnable filters.** After adaptive filtering, Flames uses two learnable linear filters: a global filter $\mathcal{M}(X_F)_G$ from the original frequency $X_F$; a local filter $\mathcal{M}(X^{high}_F)_L$ from high frequency. Each filter is tailored to match the corresponding frequency $X_F$, and $X^{high}_F$. The process is as follows:

$$X_G = \mathcal{M}(X_F)_G \odot X_F \tag{3}$$
$$X^{high}_L = \mathcal{M}(X^{high}_F)_L \odot X^{high}_F \tag{4}$$

As shown in Figure 1, we devise three variants of filter selection modules (FSM) using Fast Fourier Transformation (FFT) as filters: (a) a $1 \times 1$ convolution layer (linear). (b) Only use the ReLU activation function to dynamically select the frequency pattern. (c) Two stacks of $1 \times 1$ convolution layers and GELU activation function. We finally chose scheme (c) as our filter. Ablation experiments demonstrate the effectiveness of (c) compared to (a) frequency without selection and (b) ReLU-based selection. The three filtered spectra are then integrated as:

$$X_{mixed} = X_G + X^{high}_L \tag{5}$$

The $X_{mixed}$ is the global and local frequency mixing of $X_F$. Multiplication operation is mathematically equivalent to the dynamic circular convolutional process.

(4) **Inverse Fourier Transform.** Convert data from the frequency domain back to the time domain using inverse FFT (IFFT): $X^T = \mathcal{F}^{-1}[X_{mixed}] \in \mathcal{R}^{C \times d}$, where $\mathcal{F}^{-1}(\cdot)$ represents IFFT, ensuring the combined features are consistent with the original time series data.

During IFFT, we obtain a filtered version of the discretization step size $\Delta$, denoted $\Delta_F$. The filtered $\Delta_F$ replaces the $\Delta$ in the original Mamba, and can better capture relevant and periodic information in the presence of noise.

$$\begin{aligned} \overline{\mathbf{A}} &= \exp(\Delta_F \mathbf{A}), \\ \overline{\mathbf{B}} &= (\boldsymbol{\Delta}_F \mathbf{A})^{-1}(\exp(\Delta_F \mathbf{A}) - \mathbf{I}) \cdot \Delta_F \mathbf{B} \end{aligned} \tag{6}$$

Based on $X^T$ and the output $X'$, we can obtain the final output as follows:

$$u_i^{(1)} \leftarrow \mathbf{SSM}(\bar{\mathbf{A}}, \bar{\mathbf{B}}, \mathbf{C})(x_i^t),$$
$$u_i^{(2)} \leftarrow u_i^{(1)} \otimes \text{SiLU}(\text{Linear}(x_i')), \tag{7}$$
$$u_i \leftarrow \text{Linear}(u_i^{(2)})$$

where $x_i' \in R^C$ is $i$ time step sample. SiLU denotes the activation function. And Linear represents the linear layer. $u_i^{(1)} \in R^C$, $u_i^{(2)} \in R^C$ and $u_i \in R^C$ are three outputs.

### 3.3 MULTI-SCALE FUSION (MSF).

After performing AFFM operations on each scale and obtaining corresponding embeddings, we concatenate and feed them into the linear layer. The formula is as follows:

$$X_{MSF} \leftarrow \text{Linear}(\text{concatenate}(u_1, u_2, ..., u_M)) \tag{8}$$

where M is the number of time scales.

### 3.4 RATIONALITY OF INVERSE LAPLACE TRANSFORM.

Time series often exhibit transient dynamics, such as short-term fluctuations and abrupt changes, hindering model performance. Meanwhile, capturing long-term periodic patterns remains challenging. The Laplace transform, widely used in various domains (Holt et al., 2022), effectively analyzes exponential decay and frequency characteristics in time series. To better model both transient and steady-state behavior, we apply the inverse Laplace transform (ILT) to extract short-term dynamics and long-term trends in non-stationary sequences. Modeling them separately enhances both prediction accuracy and interpretability. The inverse Laplace transform is defined as:

$$f(t) = \mathcal{L}^{-1}\{F(s)\} = \lim_{T \to \infty} \int_{\gamma - iT}^{\gamma + iT} e^{st} F(s) \, ds \tag{9}$$

where $f(t)$ is the time-domain function, $s$ is a complex variable, and $F(s)$ is the Laplace-domain representation. Since analytical solutions may not exist for general functions, we follow (Cao et al.) and learn an operator that maps past time segments $u(t)$, $t \in [t_0, t_1]$ to future segments $u(t)$, $t \in [t_1, t_2]$ using the convolutional form:

$$u(t) = (\kappa(\phi) * v)(t) = \int_D \kappa_\phi(t - \tau) v(\tau) \, d\tau \tag{10}$$

where $\kappa$ is an integral kernel transformation. Imposing $\kappa_\phi(t, \tau) = \kappa_\phi(t - \tau)$, the Laplace domain:

$$U(s) = K_\phi(s) V(s) \tag{11}$$

where $K_\phi(s) = \mathcal{L}\{\kappa_\phi(t)\}$ and $V(s) = \mathcal{L}\{v(t)\}$, $U(s) = \mathcal{L}\{u(t)\}$.

Based on the Residue Theorem in complex analysis, the poles (singularities) in the complex plane determine its behavior in the original space. We assume $K_\phi(s) = \sum_{n=1}^{N} \frac{\beta_n}{s - \mu_n}$, where $\beta_n$ and $\mu_n$ are learnable kernel parameters. Applying a Fourier series expansion $v(t) = \sum_{l=-\infty}^{\infty} \alpha_l \exp(i\omega_l t)$, which results in $V(s) = \sum_{l=-\infty}^{\infty} \frac{\alpha_l}{s - i\omega_l}$. Mapping back into the original space:

$$u(t) = \sum_{n=1}^{N} \gamma_n \exp(\mu_n t) + \sum_{l=-\infty}^{\infty} \lambda_l \exp(i\omega_l t) \tag{12}$$

$i = \sqrt{-1}$ is imaginary, $\omega_l$ are frequencies by decomposing $v(t)$ via Fourier series, and $\gamma_n$, $\lambda_l$ are parameters from $\beta_n$, $\mu_n$, $\omega_l$ and $\alpha_l$, the former two depending on the kernel $\kappa(\phi)$, and the latter two

depending on $v(t)$. $u(t)$ is a complex number, whose real part and imagery part represent decaying and periodic behaviors. Details can see (Cao et al.). To improve efficiency and stability, we truncate the series and directly parameterize:

$$u(t) = \sum_{n=1}^{M} A_n e^{-\sigma_n t} \cos(\omega_n t + \phi_n) \tag{13}$$

where $A_n$ is amplitudes, $\sigma_n$ is the decay rate, $\omega_n$ is the frequency, and $\phi_n$ is phase, $n = 1, 2, ...$. In our work, we directly parameterize these parameters as learnable functions from historical time series via prior layers, improving the efficiency and stability of the model. Eq.13 explicitly captures transient dynamics (via $e^{-\sigma_n t}$) and periodic patterns (via $\cos(\omega_n t)$) to recognize fine-grained patterns and long-term trends.

# 4 EXPERIMENTS

## 4.1 EXPERIMENTAL SETUP

**Datasets**. We evaluate Affirm on 8 benchmarks: 4 ETTs (ETTh1, ETTh2, ETTm1, ETTm2), Electricity (ECL), Exchange, Traffic, and Weather (Wu et al., 2021). Details are in Appendix.

**Baselines**. We assessed our Flames by comparing it against ten baselines, including: 1) Transformers, such as iTransformer (Liu et al., 2024b), PatchTST (Nie et al., 2023), Crossformer (Zhang & Yan, 2023), Scaleformer (Shabani et al., 2022); 2) MLPs: Dlinear (Zeng et al., 2023), Rlinear (Li et al., 2023b), TimeMixer (Wang et al., 2024b), TiDE(Das et al.); 3) GNNs: MSGNet (Cai et al., 2024a); 4) Mambas: DTMamba(Wu et al.).

**Implementation Details** are detailed in Appendix.

## 4.2 RESULTS

We conduct a comprehensive evaluation to assess model performance with forecasting lengths $T \in \{96, 192, 336, 720\}$ using MSE and MAE. Table 1 presents our Flames performs outstandingly in most cases, with an average MSE reduction of 9.7% compared to TimeMixer and approximately 12.0% compared to PatchTST. Our Flames performs particularly well in LTSF scenarios. For instance, on the small-scale dataset ETTh1, when the prediction length equals or exceeds 336, Flames outperforms TimeMixer and PatchTST by 6.9% and 16.2% on MSE; on the large-scale dataset Traffic (862 dim), Flames surpasses TimeMixer and PatchTST by 17.9% and 22.8% on MSE. In multiscale modeling, Flames demonstrates competitive performance against MSGNet and Scaleformer, reducing the average MSE by 17.0% and 20.7% on Electricity. Besides, Flames achieves 59 wins out of 80 tests under different metrics and conditions, underscoring its versatility and effectiveness.

## 4.3 ABLATION STUDY

**Each Component of Flames.** To assess the contribution of each component in Flames, we conduct ablation studies shown in Table 2. Removing the multi-scale structure or inverse Laplace transform (ILT) significantly degrades performance, with the absence of the former a more pronounced effect. Specifically, on ETTm1 and Weather, removing multi-scale increases MSE by 6.7% and 7.7%, Excluding ILT increases MSE by 6.0% and 6.6%, indicating its critical role in time transformation and transient dynamics modeling. Removing the filter module results the increases in MSE of 5.2% and 5.8%, and MAE of 2.2% and 4.4%. We also examine the impact of removing the local adaptive component in AFFM, keeping only the global filter. The observed performance drop highlights the importance of local filtering for noise suppression. Additionally, while pre-training brings only a slight improvement, it still contributes to better overall prediction accuracy.

**Filter Selection.** To evaluate the effectiveness of FSM, we compare different variants in Figure 1 on ETTm1 and Weather, detailed in Table 3. A single $1 \times 1$ convolutional (linear) version (FSM(a)) performs the worst. Adding ReLU (FSM(b)) improves performance by partially suppressing high-frequency noise. FSM(c), which includes an extra linear layer and GELU activation, achieves the best results by enhancing nonlinearity and frequency selectivity.

Table 1: Unified hyperparameter results for the long-term forecasting task. Avg is averaged from all four prediction lengths $\in (96, 192, 336, 720)$. Count is the number of the best results. **Bold**: the best result, underlined: the second best result.

| Methods | | Flames | | MSGNet | | TimeMixer | | Scaleformer | | TimesNet | | TiDE | | DTMamba | | TimeMachine | | iTransformer | | PatchTST | | SST | | Dlinear | |
|---|---|---|---|---|---|---|---|---|---|---|---|---|---|---|---|---|---|---|---|---|---|---|---|---|---|
| Metrics | | MSE | MAE | MSE | MAE | MSE | MAE | MSE | MAE | MSE | MAE | MSE | MAE | MSE | MAE | MSE | MAE | MSE | MAE | MSE | MAE | MSE | MAE | MSE | MAE |
| ETTh1 | 96 | 0.392 | 0.410 | 0.390 | 0.411 | 0.375 | 0.400 | 0.404 | 0.441 | 0.384 | 0.402 | 0.479 | 0.464 | 0.386 | 0.399 | 0.389 | 0.402 | 0.386 | 0.405 | 0.460 | 0.447 | 0.381 | 0.405 | 0.397 | 0.412 |
| | 192 | 0.422 | 0.428 | 0.442 | 0.442 | 0.429 | 0.421 | 0.438 | 0.461 | 0.436 | 0.429 | 0.525 | 0.492 | 0.426 | 0.424 | 0.435 | 0.440 | 0.441 | 0.436 | 0.477 | 0.429 | 0.430 | 0.434 | 0.446 | 0.441 |
| | 336 | 0.462 | 0.441 | 0.480 | 0.468 | 0.484 | 0.458 | 0.464 | 0.477 | 0.491 | 0.469 | 0.565 | 0.515 | 0.480 | 0.450 | 0.450 | 0.448 | 0.487 | 0.458 | 0.546 | 0.496 | 0.443 | 0.446 | 0.489 | 0.467 |
| | 720 | 0.452 | 0.456 | 0.494 | 0.488 | 0.498 | 0.482 | 0.507 | 0.516 | 0.521 | 0.500 | 0.594 | 0.558 | 0.484 | 0.470 | 0.480 | 0.465 | 0.503 | 0.491 | 0.544 | 0.517 | 0.502 | 0.501 | 0.513 | 0.510 |
| | Avg. | 0.432 | 0.434 | 0.452 | 0.452 | 0.447 | 0.440 | 0.453 | 0.474 | 0.458 | 0.450 | 0.541 | 0.507 | 0.444 | 0.435 | 0.439 | 0.439 | 0.454 | 0.448 | 0.516 | 0.484 | 0.439 | 0.447 | 0.461 | 0.457 |
| ETTh2 | 96 | 0.301 | 0.354 | 0.328 | 0.371 | 0.289 | 0.341 | 0.335 | 0.385 | 0.340 | 0.374 | 0.400 | 0.440 | 0.290 | 0.340 | 0.230 | 0.349 | 0.297 | 0.349 | 0.308 | 0.355 | 0.291 | 0.346 | 0.340 | 0.394 |
| | 192 | 0.365 | 0.394 | 0.402 | 0.414 | 0.372 | 0.392 | 0.455 | 0.451 | 0.402 | 0.414 | 0.528 | 0.509 | 0.366 | 0.392 | 0.371 | 0.400 | 0.380 | 0.400 | 0.393 | 0.405 | 0.369 | 0.397 | 0.482 | 0.479 |
| | 336 | 0.371 | 0.407 | 0.435 | 0.443 | 0.386 | 0.414 | 0.477 | 0.479 | 0.452 | 0.452 | 0.643 | 0.571 | 0.380 | 0.409 | 0.402 | 0.449 | 0.428 | 0.432 | 0.427 | 0.436 | 0.374 | 0.414 | 0.591 | 0.541 |
| | 720 | 0.411 | 0.430 | 0.417 | 0.441 | 0.412 | 0.434 | 0.467 | 0.490 | 0.462 | 0.468 | 0.874 | 0.679 | 0.416 | 0.437 | 0.425 | 0.438 | 0.427 | 0.445 | 0.436 | 0.450 | 0.419 | 0.447 | 0.839 | 0.661 |
| | Avg. | 0.362 | 0.396 | 0.395 | 0.437 | 0.364 | 0.395 | 0.434 | 0.451 | 0.414 | 0.427 | 0.611 | 0.550 | 0.363 | 0.395 | 0.357 | 0.409 | 0.383 | 0.407 | 0.391 | 0.411 | 0.363 | 0.401 | 0.563 | 0.519 |
| ETTm1 | 96 | 0.293 | 0.356 | 0.319 | 0.366 | 0.320 | 0.357 | 0.392 | 0.415 | 0.338 | 0.375 | 0.364 | 0.387 | 0.325 | 0.360 | 0.312 | 0.371 | 0.334 | 0.368 | 0.352 | 0.374 | 0.298 | 0.358 | 0.346 | 0.374 |
| | 192 | 0.344 | 0.378 | 0.376 | 0.397 | 0.361 | 0.381 | 0.437 | 0.451 | 0.374 | 0.387 | 0.398 | 0.404 | 0.375 | 0.386 | 0.365 | 0.409 | 0.377 | 0.391 | 0.374 | 0.387 | 0.347 | 0.381 | 0.382 | 0.391 |
| | 336 | 0.372 | 0.402 | 0.417 | 0.422 | 0.390 | 0.404 | 0.499 | 0.478 | 0.410 | 0.411 | 0.428 | 0.425 | 0.396 | 0.405 | 0.421 | 0.410 | 0.426 | 0.420 | 0.421 | 0.414 | 0.374 | 0.397 | 0.415 | 0.415 |
| | 720 | 0.439 | 0.440 | 0.481 | 0.458 | 0.454 | 0.441 | 0.584 | 0.536 | 0.478 | 0.450 | 0.487 | 0.461 | 0.454 | 0.442 | 0.496 | 0.437 | 0.491 | 0.459 | 0.462 | 0.449 | 0.429 | 0.428 | 0.473 | 0.451 |
| | Avg. | 0.362 | 0.394 | 0.398 | 0.411 | 0.381 | 0.395 | 0.478 | 0.470 | 0.400 | 0.406 | 0.419 | 0.419 | 0.388 | 0.399 | 0.399 | 0.407 | 0.407 | 0.410 | 0.406 | 0.407 | 0.362 | 0.391 | 0.404 | 0.408 |
| ETTm2 | 96 | 0.185 | 0.275 | 0.177 | 0.262 | 0.175 | 0.258 | 0.182 | 0.276 | 0.187 | 0.267 | 0.207 | 0.305 | 0.177 | 0.259 | 0.185 | 0.290 | 0.180 | 0.264 | 0.183 | 0.270 | 0.176 | 0.264 | 0.193 | 0.293 |
| | 192 | 0.239 | 0.305 | 0.247 | 0.307 | 0.237 | 0.299 | 0.252 | 0.319 | 0.249 | 0.309 | 0.290 | 0.364 | 0.240 | 0.300 | 0.292 | 0.309 | 0.250 | 0.309 | 0.255 | 0.314 | 0.231 | 0.303 | 0.284 | 0.361 |
| | 336 | 0.294 | 0.338 | 0.312 | 0.346 | 0.298 | 0.340 | 0.335 | 0.372 | 0.321 | 0.351 | 0.377 | 0.422 | 0.310 | 0.345 | 0.321 | 0.367 | 0.311 | 0.348 | 0.309 | 0.347 | 0.290 | 0.339 | 0.382 | 0.429 |
| | 720 | 0.381 | 0.392 | 0.414 | 0.403 | 0.391 | 0.396 | 0.460 | 0.446 | 0.408 | 0.403 | 0.558 | 0.524 | 0.395 | 0.394 | 0.401 | 0.401 | 0.412 | 0.407 | 0.412 | 0.407 | 0.388 | 0.398 | 0.558 | 0.525 |
| | Avg. | 0.275 | 0.328 | 0.289 | 0.330 | 0.275 | 0.323 | 0.307 | 0.353 | 0.291 | 0.333 | 0.358 | 0.404 | 0.281 | 0.325 | 0.300 | 0.342 | 0.288 | 0.332 | 0.290 | 0.334 | 0.271 | 0.326 | 0.354 | 0.402 |
| Electricity | 96 | 0.128 | 0.222 | 0.165 | 0.274 | 0.153 | 0.247 | 0.168 | 0.297 | 0.168 | 0.272 | 0.237 | 0.329 | 0.166 | 0.256 | 0.156 | 0.240 | 0.148 | 0.240 | 0.190 | 0.296 | 0.141 | 0.205 | 0.210 | 0.302 |
| | 192 | 0.151 | 0.242 | 0.184 | 0.292 | 0.166 | 0.256 | 0.188 | 0.300 | 0.184 | 0.289 | 0.236 | 0.330 | 0.178 | 0.268 | 0.161 | 0.268 | 0.162 | 0.253 | 0.199 | 0.304 | 0.196 | 0.244 | 0.210 | 0.305 |
| | 336 | 0.171 | 0.268 | 0.195 | 0.302 | 0.185 | 0.277 | 0.210 | 0.324 | 0.198 | 0.300 | 0.249 | 0.344 | 0.197 | 0.289 | 0.195 | 0.272 | 0.178 | 0.272 | 0.271 | 0.319 | 0.246 | 0.283 | 0.223 | 0.319 |
| | 720 | 0.195 | 0.293 | 0.231 | 0.332 | 0.225 | 0.310 | 0.232 | 0.339 | 0.220 | 0.320 | 0.284 | 0.373 | 0.243 | 0.326 | 0.231 | 0.307 | 0.225 | 0.317 | 0.258 | 0.352 | 0.314 | 0.334 | 0.258 | 0.350 |
| | Avg. | 0.161 | 0.256 | 0.194 | 0.300 | 0.182 | 0.272 | 0.203 | 0.315 | 0.192 | 0.295 | 0.251 | 0.344 | 0.196 | 0.285 | 0.186 | 0.272 | 0.178 | 0.270 | 0.216 | 0.318 | 0.224 | 0.267 | 0.225 | 0.319 |
| Exchange | 96 | 0.089 | 0.204 | 0.102 | 0.230 | 0.091 | 0.215 | 0.109 | 0.240 | 0.107 | 0.234 | 0.094 | 0.218 | 0.083 | 0.201 | 0.087 | 0.206 | 0.086 | 0.206 | 0.088 | 0.205 | 0.081 | 0.202 | 0.088 | 0.218 |
| | 192 | 0.175 | 0.297 | 0.195 | 0.317 | 0.197 | 0.318 | 0.241 | 0.353 | 0.226 | 0.344 | 0.184 | 0.307 | 0.173 | 0.295 | 0.180 | 0.300 | 0.177 | 0.299 | 0.176 | 0.299 | 0.176 | 0.298 | 0.176 | 0.315 |
| | 336 | 0.327 | 0.414 | 0.359 | 0.436 | 0.416 | 0.472 | 0.471 | 0.508 | 0.367 | 0.448 | 0.349 | 0.431 | 0.346 | 0.427 | 0.346 | 0.423 | 0.331 | 0.417 | 0.301 | 0.397 | 0.341 | 0.420 | 0.313 | 0.427 |
| | 720 | 0.878 | 0.696 | 0.940 | 0.738 | 0.968 | 0.725 | 1.259 | 0.865 | 0.964 | 0.746 | 0.852 | 0.698 | 0.868 | 0.698 | 0.943 | 0.720 | 0.847 | 0.691 | 0.901 | 0.714 | 0.889 | 0.704 | 0.839 | 0.695 |
| | Avg. | 0.367 | 0.403 | 0.399 | 0.430 | 0.418 | 0.433 | 0.520 | 0.492 | 0.416 | 0.443 | 0.370 | 0.413 | 0.368 | 0.405 | 0.389 | 0.412 | 0.360 | 0.403 | 0.367 | 0.404 | 0.372 | 0.406 | 0.354 | 0.414 |
| Traffic | 96 | 0.358 | 0.251 | 0.605 | 0.344 | 0.462 | 0.285 | 0.564 | 0.351 | 0.593 | 0.321 | 0.805 | 0.493 | 0.487 | 0.317 | 0.398 | 0.274 | 0.395 | 0.268 | 0.526 | 0.347 | 0.367 | 0.257 | 0.650 | 0.396 |
| | 192 | 0.382 | 0.268 | 0.613 | 0.359 | 0.473 | 0.296 | 0.570 | 0.349 | 0.617 | 0.336 | 0.756 | 0.474 | 0.498 | 0.325 | 0.393 | 0.282 | 0.417 | 0.276 | 0.522 | 0.332 | 0.385 | 0.266 | 0.598 | 0.370 |
| | 336 | 0.398 | 0.277 | 0.642 | 0.376 | 0.498 | 0.296 | 0.576 | 0.349 | 0.629 | 0.336 | 0.762 | 0.477 | 0.511 | 0.334 | 0.443 | 0.368 | 0.433 | 0.283 | 0.517 | 0.334 | 0.401 | 0.275 | 0.605 | 0.373 |
| | 720 | 0.427 | 0.296 | 0.702 | 0.402 | 0.506 | 0.313 | 0.602 | 0.360 | 0.640 | 0.350 | 0.719 | 0.449 | 0.533 | 0.326 | 0.470 | 0.309 | 0.467 | 0.302 | 0.552 | 0.352 | 0.445 | 0.302 | 0.645 | 0.394 |
| | Avg. | 0.391 | 0.273 | 0.641 | 0.370 | 0.484 | 0.297 | 0.578 | 0.352 | 0.620 | 0.336 | 0.760 | 0.473 | 0.507 | 0.326 | 0.426 | 0.308 | 0.428 | 0.282 | 0.529 | 0.341 | 0.400 | 0.275 | 0.625 | 0.383 |
| Weather | 96 | 0.152 | 0.203 | 0.163 | 0.212 | 0.163 | 0.209 | 0.220 | 0.289 | 0.172 | 0.220 | 0.202 | 0.261 | 0.171 | 0.218 | 0.174 | 0.218 | 0.174 | 0.214 | 0.160 | 0.204 | 0.153 | 0.205 | 0.195 | 0.252 |
| | 192 | 0.196 | 0.231 | 0.212 | 0.254 | 0.208 | 0.250 | 0.341 | 0.385 | 0.219 | 0.261 | 0.242 | 0.298 | 0.220 | 0.257 | 0.200 | 0.258 | 0.221 | 0.254 | 0.204 | 0.245 | 0.196 | 0.244 | 0.237 | 0.295 |
| | 336 | 0.246 | 0.285 | 0.272 | 0.299 | 0.251 | 0.287 | 0.463 | 0.455 | 0.280 | 0.306 | 0.287 | 0.335 | 0.274 | 0.296 | 0.280 | 0.299 | 0.278 | 0.296 | 0.257 | 0.285 | 0.246 | 0.283 | 0.282 | 0.331 |
| | 720 | 0.315 | 0.335 | 0.350 | 0.348 | 0.339 | 0.341 | 0.682 | 0.565 | 0.365 | 0.359 | 0.351 | 0.386 | 0.349 | 0.346 | 0.352 | 0.359 | 0.358 | 0.349 | 0.329 | 0.338 | 0.314 | 0.334 | 0.345 | 0.382 |
| | Avg. | 0.227 | 0.263 | 0.249 | 0.278 | 0.240 | 0.271 | 0.427 | 0.424 | 0.259 | 0.287 | 0.271 | 0.287 | 0.254 | 0.279 | 0.252 | 0.284 | 0.258 | 0.278 | 0.238 | 0.268 | 0.227 | 0.267 | 0.265 | 0.315 |
| 1st Count | | 47 | | 0 | | 9 | | 0 | | 0 | | 0 | | 5 | | 2 | | 1 | | 2 | | 12 | | 2 | |

Table 2: Ablation study of each component in Flames. Results are averaged from {96,192,336,720}.

| Variants | ETTm1 (avg.) | | Weather (avg.) | |
|---|---|---|---|---|
| | MSE | MAE | MSE | MAE |
| w/o Multi-scale | 0.388 | 0.413 | 0.246 | 0.286 |
| w/o ILT | 0.385 | 0.409 | 0.243 | 0.280 |
| w/o Filter | 0.382 | 0.406 | 0.241 | 0.275 |
| w/o AHFD | 0.376 | 0.403 | 0.234 | 0.272 |
| w/o pretraining | 0.370 | 0.400 | 0.231 | 0.270 |
| **Flames** | **0.362** | **0.394** | **0.227** | **0.263** |

**Varying Look-back Window.** To assess the impact of look-back window length on performance, we compare Flames with six baselines on three large-scale datasets: Electricity, Traffic, and Weather across window sizes $\in \{24, 48, 96192, 336, 720\}$. While longer windows improve forecasting by expanding the receptive field, most Transformer-based models (e.g., Informer, Autoformer)

Table 3: Ablation studies on different variants of FSM illustrated in Figure 1.

| Variants | ETTm1 (avg.) | | Weather (avg.) | |
|---|---|---|---|---|
| | MSE | MAE | MSE | MAE |
| FSM (a) | 0.369 | 0.404 | 0.241 | 0.272 |
| FSM (b) | 0.366 | 0.401 | 0.234 | 0.266 |
| **FSM (c)** | **0.362** | **0.394** | **0.227** | **0.263** |

show limited gains, likely due to weak temporal modeling or unstable training in Figure 2. In contrast, PatchTST, TimeMixer, and Flames consistently benefit from longer windows, with Flames achieving best and most stable performance across all settings, demonstrating strong generalization in both short- and long-term forecasting.

## 4.4 ROBUSTNESS ANALYSIS

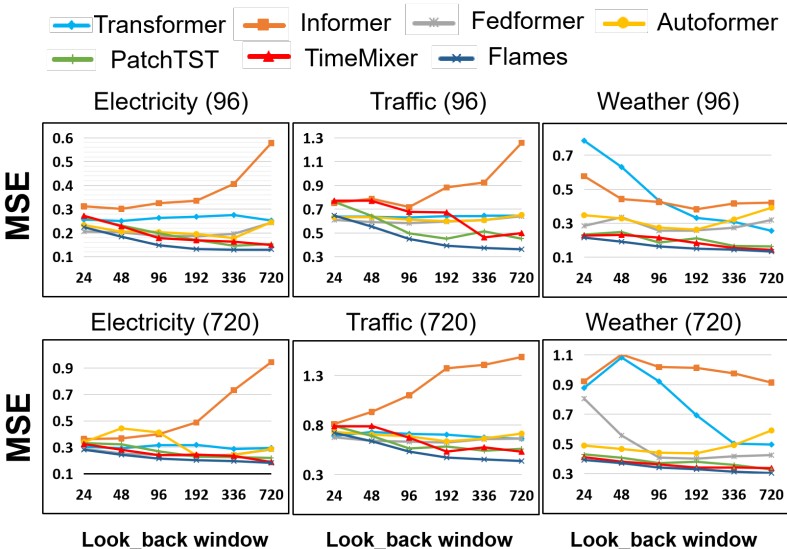

Figure 2: MSE on 3 large datasets with varying look-back window sizes.

Figure 3 compares the performance of Transformer, PatchTST, Flames w/o, and Flames under varying levels of Gaussian noise on the Weather and ETTh1 datasets. As noise increases, Transformer performance drops sharply, while PatchTST is more noise-sensitive on Weather than ETTh1. In contrast, both Flames variants maintain stable performance, with Flames (with filters) showing the strongest robustness. Notably, Flames exhibits signifi-

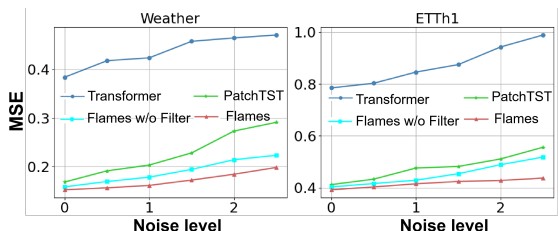

Figure 3: Robustness against noise on Weather and ETTh1.

cantly less degradation under noise compared to other methods, highlighting the benefit of adaptive filtering. Even without the filter, Flames outperforms Transformer and PatchTST, underscoring the value of the Laplace transform and multi-scale design. In comparison, Transformer-based models show poor noise tolerance.

## 4.5 SCALABILITY ANALYSIS

To evaluate the scalability of Flames, we compare it with TimeMixer. Figure 4 illustrates their performance on ETTm1 across varying data sizes and network depths. Flames consistently outperforms TimeMixer across all settings and confirms greater stability. As the training data increases, both models exhibit slight fluctuations, but TimeMixer's performance degrades more noticeably, particularly under low-resource scenarios with only 1% data volume, where deeper networks lead to significant performance drops. This suggests TimeMixer is more susceptible to overfitting or optimization challenges as model complexity increases. This phenomenon may stem from the architecture design of TimeMixer - its MLP module may lack effective inductive bias when deepening the model, leading to increased optimization difficulty and hindering feature extraction. In contrast, Flames can better adapt to various scenarios with various data ratios and layer counts, and maintain stability or even further improve performance, highlighting its superior generalization and scalability.

## 4.6 ANALYSIS ON NUMBER OF SCALES

We explore the impact of the number of scales (M) under varying prediction lengths, as shown in Figure 5. The results show that performance initially improves with increasing M. While M=4 yields the best results at lengths 192 and 336, it introduces higher computational overhead. Overall, M=3 offers a better trade-off between performance and efficiency.

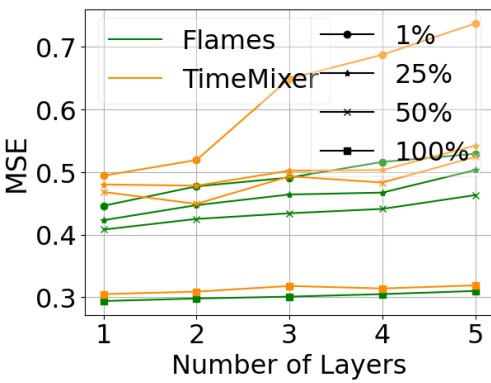 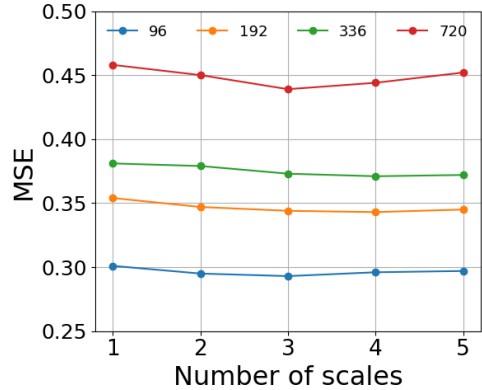

Figure 4: Scaling analysis on layer counts and data ratios on ETTm1.

Figure 5: Analysis scales number M on ETTm1.

## 4.7 COMPLEXITY ANALYSIS

We evaluate model computational efficiency on high-dimensional ECL, focusing on trainable parameters, MACs, inference time, and MSE over a 96 look-back window and a 720 prediction length. Table 4 highlights Flames's strong balance between efficiency and performance. Transformer, Informer, Autoformer, FEDformer, and FiLM have 13.61M–20.68M parameters and 3.93G–5.97G MACs, 26.8ms -123.0 inference time, yet their MSEs are much higher than Flames. Compared to lightweight PatchTST and TSLAnet, Flames maintains similar parameter sizes but reduces MACs by 25.8% and 50.7%, inference time by 15.2% and 22.2%, and MSE by 24.4% and 4.9%, respectively. Notably, compared to TimesNet, Flames cuts both parameters and MACs by over 99%, while also achieving an 11.4% lower MSE. Results show Flames delivers superior accuracy with lower computational cost, making it a compelling lightweight solution for resource-constrained scenarios.

Table 4: Number of training parameters, MACs, inference time, and MSE of TSF models under look-back window=96 and forecasting horizon=720 on the large Electricity dataset.

| Method | MACs | Parameters | Infer.Time | MSE |
|---|---|---|---|---|
| Transformer | 4.03G | 13.61M | 26.8ms | 0.491 |
| Informer | 3.93G | 14.38M | 49.3ms | 0.399 |
| Autoformer | 4.41G | 14.91M | 164.1ms | 0.412 |
| FEDfomrer | 4.41G | 20.68M | 40.5ms | 0.264 |
| PatchTST | 5.07G | 1.5M | 3.3ms | 0.258 |
| FiLM | 5.97G | 14.91M | 123.0ms | 0.268 |
| TimesNet | 1226.49G | 301.7M | N/A | 0.220 |
| TSLANet | 7.62G | **1.4M** | 3.6ms | 0.205 |
| **Flames** | **3.76G** | 1.5M | **2.8ms** | **0.195** |

## 5 CONCLUSION

In this paper, we propose Flames, a novel lightweight model for TSF that embeds adaptive Fourier filters into Mamba, offering a compelling alternative to Transformers. It performs multi-scale periodic modeling by integrating adaptive global-local Fourier filters. Local neural operator applies learnable thresholds to filter out noise, and dropout regularization enhances parameter selectivity and mitigates overfitting. Moreover, the Laplace transform is introduced to capture transient dynamics within time series. Extensive experiments show Flames's strong accuracy–efficiency trade-offs across noise levels and data sizes. Robustness analysis confirms its ability to handle different noise intensities, while complexity analysis highlights substantial reductions in computational costs and inference time. Furthermore, an in-depth layer-wise scaling analysis reveals that Flames has superior scalability on deeper depths and smaller datasets. Overall, Flames offers an efficient and robust solution for LTSF, paving the way for its use as a strong foundational model.

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

## A  RELATED WORKS

### A.1  TIME SERIES FORECASTING

Time series forecasting used to rely on traditional VAR (Kilian & Lütkepohl, 2017) and Prophet (Taylor & Letham, 2018), yet they often struggle with the complex dynamic nature of time series. Deep learning models— CNNs, RNNs, MLPs, and Transformers—have shown strong power in capturing temporal dependencies (Lim & Zohren, 2021). CNNs (Liu et al., 2022a; Wang et al., 2023) leverage convolutional kernels along the temporal axis to extract time-dependent patterns, while RNNs (Lai et al., 2018) model sequential transitions through recurrent structures. MLPs (Zeng et al., 2023) map temporal relationships directly. For instance, DLinear (Zeng et al., 2023) uses a single-layer linear model to map temporal dependencies, while RLinear (Li et al., 2023b) captures periodic features through linear mapping, maintaining robustness across varying input lengths. Transformers (Nie et al., 2023; Liu et al., 2024b) have become dominant in TSF due to their self-attention mechanism. PatchTST (Nie et al., 2023) segments sequences into patches to preserve channel independence, iTransformer (Liu et al., 2024b) reverses the Transformer structure by treating sequences as variable markers to enhance multivariate correlation learning. However, they struggle with small datasets, easily falling into overfitting and computational inefficiencies due to notorious large parameter sizes and quadratic complexity.

To better capture complex temporal patterns, several strategies have been proposed, such as sequence decomposition and multi-period analysis. DLinear (Zeng et al., 2023) uses decomposition as preprocessing, while Autoformer (Wu et al., 2021) integrates a decomposition block to separate seasonal and trend components. TimesNet (Wu et al., 2022) applies Fourier transforms to extract multi-period components, with a modular design for handling each. Quatformer (Chen et al., 2022) introduces learnable period and phase information to depict intricate periodical patterns, and FiLM (Zhou et al., 2022a) maps series into Legendre polynomial space, where basis functions represent different periodic patterns.

In contrast to prior work, we focus on multi-scale modeling that also addresses noise and transient dynamics. While models like Pyraformer (Liu et al., 2022b) use pyramid attention, SCINet (Liu et al., 2022a) utilizes bifurcate downsampling trees, and TimeMixer (Wang et al., 2024b) employs disentangled sequences for multi-scale learning, they all overlook noise reduction and short-term fluctuations. We propose Flames, a novel Mamba-based framework that offers a promising alternative to Transformers with parallel training, linear complexity, and strong sequence modeling capabilities. Flames integrates adaptive Fourier filtering with learnable thresholds and the Laplace transform to enhance noise suppression and transient dynamics modeling. This design enables effective multi-scale periodicity modeling while balancing computational efficiency and long-term prediction accuracy.

## B  METHOD

### B.1  COMBINED OPERATOR VIEW OF FLAMES

This section provides a combined view that clarifies how they jointly define a flexible yet structured forecasting operator.

**Multi-scale decomposition.** Given an input time series $x \in \mathbb{R}^{T \times C}$, Flames constructs multi-scale inputs $\{x_m\}_{m=0}^{M}$ via the recursive downsampling

$$x_m = \text{Conv}(x_{m-1}, \text{stride} = 2), \quad m \in \{1, \ldots, M\}, \tag{14}$$

where $x_0 = x$ and $x_m \in \mathbb{R}^{\lfloor T/2^m \rfloor \times C}$. An embedding layer then projects each scale into deep features

$$X' = \text{Embed}(X) = \{x'_0, \ldots, x'_M\}, \quad x'_m \in \mathbb{R}^{\lfloor T/2^m \rfloor \times d_{\text{model}}}, \tag{15}$$

which serve as the inputs of AFFM at different resolutions.

**AFFM as a data-dependent spectral operator.** For a given scale $m$, let $x'_m$ denote the embedded sequence. AFFM first transforms $x'_m$ from the time domain to the frequency domain by applying a

1D FFT along the temporal dimension,

$$X_{F,m} = \mathcal{F}[x'_m], \tag{16}$$

where $\mathcal{F}[\cdot]$ is the FFT operator. As detailed in Sec. 3.2, adaptive high-frequency denoising and learnable global/local filters are then applied in the frequency domain:

$$X_{F,m}^{\text{high}} = X_{F,m} \odot (P > \theta_{\text{high}}), \tag{17}$$

$$X_{G,m} = \mathcal{M}(X_{F,m})_G \odot X_{F,m}, \qquad X_{L,m}^{\text{high}} = \mathcal{M}(X_{F,m}^{\text{high}})_L \odot X_{F,m}^{\text{high}}, \tag{18}$$

and the filtered spectrum at scale $m$ is obtained by

$$X_{mixed,m} = X_{G,m} + X_{L,m}^{\text{high}}. \tag{19}$$

Finally, an inverse FFT (IFFT) maps the mixed spectrum back to the time domain,

$$X_m^T = \mathcal{F}^{-1}[X_{mixed,m}], \tag{20}$$

where $\mathcal{F}^{-1}[\cdot]$ denotes IFFT and $X_m^T$ has the same temporal shape as $x'_m$. By the convolution theorem, the element-wise multiplication in the frequency domain is equivalent to a convolution in the time domain, i.e., there exists a data-dependent kernel $h_m$ such that

$$X_m^T(t) = (h_m * x'_m)(t), \tag{21}$$

where $*$ denotes convolution along the temporal dimension. Therefore, AFFM can be interpreted as learning a *spectrally shaped kernel* $h_m$ that filters out noise and emphasizes task-relevant frequency bands for each scale, rather than only applying a point-wise nonlinearity on top of Mamba states.

**Inverse Laplace head as a damped-modal parameterization.** On top of the AFFM-filtered representation, Flames employs the inverse Laplace transform (ILT) head discussed in Sec. 3.4. Eq. equation 13 parameterizes the forecast $u(t)$ as a superposition of damped sinusoidal modes,

$$u(t) = \sum_{n=1}^{M} A_n e^{-\sigma_n t} \cos(\omega_n t + \phi_n), \tag{22}$$

where $A_n$, $\sigma_n$, $\omega_n$, and $\phi_n$ are learnable functions of the historical sequence. For each scale $m$, we instantiate this parameterization with its own parameters $\{A_{m,n}, \sigma_{m,n}, \omega_{m,n}, \phi_{m,n}\}_{n=1}^{M}$ that are predicted from $X_m^T$:

$$u^{(m)}(t) = \sum_{n=1}^{M} A_{m,n} e^{-\sigma_{m,n} t} \cos(\omega_{m,n} t + \phi_{m,n}), \tag{23}$$

so that small $\sigma_{m,n}$ encode long-term periodic trends and large $\sigma_{m,n}$ capture short-lived transient dynamics at scale $m$.

**Overall operator and special cases.** Combining the two views above, the contribution of scale $m$ to the prediction can be written as

$$u^{(m)}(t) = \sum_{n=1}^{M} A_{m,n} (h_m * x'_m)(t) e^{-\sigma_{m,n} t} \cos(\omega_{m,n} t + \phi_{m,n}), \tag{24}$$

where $h_m$ is the AFFM-induced kernel for scale $m$ and $x'_m$ is the embedded multi-scale input. The outputs $\{u^{(m)}\}_{m=0}^{M}$ from all scales are then fused by the multi-scale fusion module (Sec. 3.3), which concatenates the scale-wise representations and applies a linear projection to obtain the final forecast.

From this unified operator view, Flames can be seen as learning a dictionary of multi-scale damped sinusoidal kernels with data-dependent spectral selection. It strictly generalizes several widely used architectures: (i) when the frequency response degenerates to identity (i.e., $h_m$ becomes a delta kernel), AFFM reduces to the vanilla Mamba update and Flames becomes a Mamba-based forecaster with an ILT head; (ii) when the ILT head degenerates to a linear readout (e.g., fixing $\sigma_{m,n} = 0$ and $\omega_{m,n} = 0$), Flames reduces to an AFFM-enhanced Mamba model; and (iii) when the Mamba state update is fixed and only the spectral filters are learned, Flames recovers a pure Fourier-filter model. This analysis shows that Flames is more than a simple stacking of existing tools: it defines a coherent, interpretable operator tailored to simultaneously handle multi-scale periodicity, noise robustness, and transient dynamics in time series forecasting.

## B.2 MORE MOTIVATION

In this section, we provide additional motivation for the three key challenges highlighted in the introduction—(i) multi-scale periodicity, (ii) data noise, and (iii) transient dynamics—and explain why they are particularly critical for long-term time series forecasting (LTSF).

### B.2.1 MULTI-SCALE PERIODICITY

Real-world time series rarely evolve on a single time scale. For example, electricity and traffic series typically contain daily and weekly cycles, superimposed with slower monthly or seasonal trends; weather data show diurnal, synoptic, and annual periodicities; financial and sensor data often exhibit both high-frequency fluctuations and low-frequency drifts. These patterns coexist and interact across different resolutions, and the relevant scale can change over time and across variables. Effective modeling therefore requires capturing both fine-grained (microscopic) and coarse-grained (macroscopic) temporal patterns, instead of relying on a single fixed resolution Hu et al. (2025); Wang et al. (2025).

Recent multi-scale designs such as Scaleformer Shabani et al. (2022), MSGNet Cai et al. (2024a) and TimeMixer Wang et al. (2024b) introduce hierarchical modules or multi-resolution mixing, which confirms that explicitly modeling multiple scales is beneficial. However, the multi-scale research based on Mamba is rarely explored, which makes it difficult to simultaneously model fast local variations and slow global trends, especially when the prediction horizon is long.

These observations motivate our multi-scale design in Flames: we progressively downsample the input series to construct a hierarchy $\{x_m\}_{m=0}^{M}$ with decreasing temporal resolutions, and then encode each scale with a shared Adaptive Fourier Filter Mamba (AFFM) block. The multi-scale fusion module aggregates the representations across scales, allowing the model to align and combine short-term and long-term patterns in a lightweight way. As shown by the ablation studies, removing the multi-scale component leads to a notable degradation in performance, which empirically validates the importance of this design.

### B.2.2 DATA NOISE AND ROBUSTNESS

Time series collected from IoT devices, industrial systems, medical monitoring or financial markets are inevitably contaminated by various sources of noise: sensor errors, communication glitches, missing values filled by ad-hoc imputation, sudden distribution shifts, or exogenous shocks Li et al. (2023a). Such noise often manifests as high-frequency components or irregular spikes superimposed on the underlying structured signal Eldele et al. (2024). If these noisy components are not properly handled, models tend to overfit them and suffer from unstable predictions and poor generalization, especially in long-term horizons.

Different backbones are affected by noise in different ways. Self-attention in Transformers can assign large weights to noisy positions and thus amplify spurious correlations; MLP- and Linear-based models, while computationally efficient, lack explicit mechanisms to separate informative patterns from noise, so their performance degrades noticeably on high-dimensional or noisy datasets. State Space Models, including Mamba, are more parameter-efficient, but the selective scan mechanism forwards all input signals through the step size $\Delta$ without explicit denoising. Consequently, noisy high-frequency components can still be propagated through the state evolution, corrupting long-range dependency modeling.

These issues motivate the design of our Adaptive Fourier Filter Mamba (AFFM). Instead of directly operating in the time domain, we transform latent features into the frequency domain, where different frequency bands correspond to different temporal behaviors. We introduce an Adaptive High-Frequency Denoising (AHFD) module with a learnable threshold, which suppresses irrelevant high-frequency noise while preserving informative components. Furthermore, we design global and local learnable filters to model both full-spectrum interactions and fine-scale corrections, and we fuse them via element-wise multiplication, which is mathematically equivalent to a dynamic circular convolution.

This frequency-domain perspective provides two benefits: (1) it explicitly reduces noise before the state update in Mamba, leading to more stable dynamics; and (2) it allows the model to emphasize

task-relevant frequencies at different scales, which is crucial for robust LTSF. Our robustness experiments (Section 4.4) with injected Gaussian noise on Weather and ETTh1 demonstrate that Flames degrades much more gracefully than Transformer and PatchTST, confirming that adaptive frequency filtering substantially improves noise robustness.

### B.2.3 TRANSIENT DYNAMICS AND EVENT-DRIVEN BEHAVIOR

Besides periodic and slowly varying trends, many real-world series exhibit *transient dynamics*: short-lived spikes, abrupt regime switches, event-driven responses, or rapidly decaying shocks Zhang et al. (2025). Examples include traffic jams caused by accidents Orosz et al. (2005), power load surges due to heat waves Wang & Nehrir (2007), sudden policy changes in economics Martin et al. (2020); Jalan & Ravallion (1998), or physiological responses in healthcare data Janićijević et al. (2024). These transient patterns often have strong predictive value, but they are localized in time and may not repeat regularly, so they are difficult to capture with models that primarily focus on stationary or periodic structures.

To better capture transient dynamics, we introduce an inverse Laplace transform (ILT) module on top of the multi-scale Mamba features. The Laplace transform provides a natural representation for functions composed of exponentially decaying and oscillatory components Dobaczewski et al. (1994); Hamedi-Hagh (2021); Sundararajan (2022). Following the theoretical formulation of Laplace neural operators, we parameterize the Laplace-domain kernel as a sum of rational functions whose poles correspond to decay rates and frequencies, and then map back to the time domain. In our implementation, this leads to an explicit representation of the form

$$u(t) = \sum_{n=1}^{M} A_n e^{-\sigma_n t} \cos(\omega_n t + \phi_n),$$

where $A_n$, $\sigma_n$, $\omega_n$, and $\phi_n$ are learned from data. The exponential terms model transient, quickly decaying effects, while the cosine terms capture oscillatory behavior.

By combining ILT with AFFM and multi-scale downsampling, Flames separates and jointly models: (i) long-term periodicities and trends (captured by multi-scale Fourier filters), (ii) noise components (attenuated by adaptive high-frequency filtering), and (iii) localized transients (modeled via the Laplace branch). Ablation studies show that removing the ILT branch consistently increases both MSE and MAE, indicating that explicitly modeling transient dynamics provides non-trivial gains on benchmark datasets.

### B.3 REASON FOR CHOOSING MAMBA AS BACKBONE

Our choice of Mamba is based on the following considerations: (i) Mamba offers linear-time complexity and good scalability for long sequence modeling Patro & Agneeswaran (2024); (ii) existing Mamba variants pay relatively little attention to frequency-domain modeling and transient dynamics, leaving considerable room for improvement; and (iii) our proposed combination of multi-scale design, adaptive Fourier filtering, and Laplace-based operator naturally complements the selective state-space structure of Mamba, yielding a lightweight yet effective alternative.

### B.4 PSEUDOCODE FOR MAMBA WITH ADAPTIVE FOURIER FILTER (AFF)

**Algorithm 1** summarizes how we wrap a standard Mamba-style SSM cell with the proposed Adaptive Fourier Filter (AFF). Given the input sequence $x$, we first compute the raw step sizes $\Delta = f_\Delta(x)$ as in the original Mamba block. Then AFF operates on $\Delta$ in a purely modular manner: it performs a 1D FFT along the temporal dimension, applies a learnable frequency-domain mask constructed from a global and a local $1 \times 1$ convolution, and transforms the filtered spectrum back to the time domain to obtain $\Delta_F$. Finally, $\Delta_F$ is passed to the unchanged Mamba/SSM update to produce the output sequence $y$ and the final hidden state $s_T$. Importantly, the internal SSM equations and the rest of the Mamba implementation remain intact; from an implementation perspective, wrapping a Mamba cell with AFF amounts to replacing `delta = f_Delta(x)` by `delta = AFF(f_Delta(x))`.

---

**Algorithm 1** Mamba Cell with Adaptive Fourier Filter (AFF)

---

**Require:** Input sequence $x \in \mathbb{R}^{B \times T \times d}$;
    SSM input $u \in \mathbb{R}^{B \times T \times d_u}$;
    Initial state $s_0$;
    Step-size network $f_\Delta(\cdot)$;
    Global and local $1 \times 1$ convolutions $g_{\text{glob}}, g_{\text{loc}}$;
    Base Mamba/SSM cell $\text{Mamba}(\cdot)$.
**Ensure:** Output sequence $y$, final state $s_T$.

1: **Step 1: Compute raw step sizes**
2: $\Delta \leftarrow f_\Delta(x)$          // $\Delta \in \mathbb{R}^{B \times T \times d}$
3: **Step 2: Adaptive Fourier Filter (AFF)**
4: Rearrange $\Delta$ to time–channel layout: $\Delta^{\text{td}} \leftarrow \text{reshape}(\Delta)$    // $\Delta^{\text{td}} \in \mathbb{R}^{B \times d \times T}$
5: Compute temporal FFT: $\Delta^{\text{fd}} \leftarrow \text{FFT}(\Delta^{\text{td}})$
6: Compute global and local masks in time domain: $m_{\text{glob}} \leftarrow g_{\text{glob}}(\Delta^{\text{td}}), m_{\text{loc}} \leftarrow g_{\text{loc}}(\Delta^{\text{td}})$
7: Combine and squash to $(0,1)$: $M \leftarrow \sigma(m_{\text{glob}} + m_{\text{loc}})$    // $M \in \mathbb{R}^{B \times d \times T}$
8: Apply mask in frequency domain: $\Delta^{\text{fd}}_F \leftarrow \Delta^{\text{fd}} \odot M$
9: Inverse FFT back to time domain: $\Delta^{\text{td}}_F \leftarrow \Re(\text{IFFT}(\Delta^{\text{fd}}_F))$
10: Rearrange back to original layout: $\Delta_F \leftarrow \text{reshape}(\Delta^{\text{td}}_F)$    // $\Delta_F \in \mathbb{R}^{B \times T \times d}$
11: **Step 3: Run original Mamba/SSM update with filtered step sizes**
12: $(y, s_T) \leftarrow \text{Mamba}(u, \Delta_F, s_0)$ $\{y$: output sequence; $s_T$: final hidden state$\}$
13: **return** $(y, s_T)$

---

## C EXPERIMENTS

**Datasets**. To verify the model's performance, we evaluate the performance of our proposed Flames on 8 popular, well-established benchmarks: 4 ETT datasets (ETTh1, ETTh2, ETTm1, ETTm2), Electricity, Exchange, Traffic, and Weather. The datasets have been extensively adopted for benchmarking LSTF models and are publicly available on (Wu et al., 2021), covering domains such as electricity, transportation, energy, weather, and economy. Notably, we would like to highlight 3 large datasets: Electricity, Traffic, and Weather. They have more dimensions, so the results will be more stable and less prone to overfitting than other smaller datasets. (1) **ETT** [1] comprises two granularities collected from different regions in China, containing two hourly-level datasets (ETTh1, ETTh2) and two 15-minute-level datasets (ETTm1, ETTm2). Each dataset includes six power load features and a target "oil temperature" variable from July 2016 to July 2018. (2) **Electricity** tracks the power electricity consumption of 321 clients, converted to hourly measurements. This provides insights into usage patterns and enables demand forecasting, which is crucial for optimizing power generation and distribution. (3) **Exchange Rate** features daily exchange rates of different currencies from 8 different countries against the US dollar, allowing the model to predict currency fluctuations based on historical data. (4) **Traffic** records the road occupancy rate of different sensors on the San Francisco highway, including the traffic volume of 94 interstate westbound traffic in the Twin Cities metropolitan area, which can predict the traffic flow pattern, crucial to congestion management and urban planning. (5)**Weather**[2] contains 35,136 data points from a PV power station in northwest China, 15-minute intervals from January 1, 2020, to December 31, 2020. Besides, we will publicize this data as one benchmark. The statistics details are highlighted in Table 5.

**Implementation Details**. We conduct all experiments on an NVIDIA GeForce RTX 4090 Ti GPU with 64-bit Linux 5.15.0-56-generic, with a 60/20/20 train/validation/test split for ETTs, and 70/10/20 for other datasets. Adhere to Time-Series-Library settings (Wang et al., 2024d), we use the look-back window of 336 for ETTs, 96 for Exchange and Electricity, 512 for Traffic and Weather. Additionally, we incorporated a data normalization module and a reverse instance norm (Kim et al., 2021). For each baseline, if their setup matches ours, we report the best results from their original work; otherwise, we rerun their code to obtain the results. To handle longer series in LTSF, we set M to 3 to trade off performance and efficiency.

---

[1]https://github.com/zhouhaoyi/ETDataset
[2]https://www.ncei.noaa.gov/data/local-climatological-data/

| Dataset | Dim | Dataset Size | Frequency | Domain |
|---------|-----|--------------|-----------|--------|
| ETTh1 | 7 | (8545, 2881, 2881) | Hourly | Temerature |
| ETTh2 | 7 | (8545, 2881, 2881) | Hourly | Temerature |
| ETTm1 | 7 | (34465, 11521, 11521) | 15min | Temerature |
| ETTm2 | 7 | (34465, 11521, 11521) | 15min | Temerature |
| Exchange | 8 | (5120, 665, 1422) | Daily | Economy |
| Electricity | 321 | (18317, 2633, 5261) | Hourly | Electricity |
| Traffic | 862 | (12185, 1757, 3509) | Hourly | Transportation |
| Weather | 21 | (36792, 5271, 10540) | 10min | Weather |

Table 5: Dataset detailed descriptions. Dim denotes the variate number of each dataset. Frequency represents the sampling interval of time points. The dataset size is organized in (Train, Validation, Test).

## C.1 MORE COMPARATIVE BASELINES

We compare our model with other common time series forecasting baselines, i.e. 1) Mamba-based: S-Mamba (**?**), 2) Transformer-based: Autoformer (Wu et al., 2021) , FEDformer (Zhou et al., 2022b), 3) CNN-based: SCINet (Liu et al., 2022a), MICN (Wang et al., 2023). 4) Frequency-based: FiLM (Zhou et al., 2022a) and FreTS (Yi et al., 2024b).

Table 6 shows that our proposed Flames still exhibits significant advantages in all datasets and all lengths, For instance, on Weather, the MSE of our method decreases by 9.6%, 16.2%, 14.0%, 15.3%, 9.6%, 22.3%, 32.8%, and 26.5%. Similarly, the MAE exceeds other models by 4.7%, 9.3%, 15.4%, 18.1%, 4.7%,27.5%, 31.2%, and 26.9%, which indicates the effectiveness and superiority of our framework in time series forecasting. Overall, our Affirm wins in 64 out of 80 results, and 13 second-best in the table.

## C.2 MORE MAMBA AND FREQUENCY FILTER BASELINES

Following the reviewers' suggestions, we enriched the baselines in Table 2 to cover a broader spectrum of representative architectures: (1) Mamba-based models, including FLDMamba, SiMBA, and Time-SSM, to compare different state-space modeling paradigms; (2) a frequency-filtering model, FilterNet, to examine the benefits of frequency-domain modeling; (3) a Transformer-based model, Crossformer, representing typical attention-based architectures for long-sequence modeling; and (4) a linear model, RLinear, serving as a strong and lightweight baseline. As shown in Table **??**, Flames achieves the best performance in 54 out of 80 comparison entries and remains very close to the optimal error in most of the remaining cases, across different datasets and forecasting horizons for both MSE and MAE. We can see Flames performs better in higher-dimensional and more complex d Electricity (321 dim), Traffic (862 dim), and Weather (21 dim) datasets, which indicates that Flames has better scalability and robustness on real-world multivariate time series with high dimensionality and strong noise, and can more effectively capture both temporal dependencies and cross-variable interactions. These results demonstrate that Flames enjoys a stable and pronounced overall advantage in a systematic comparison across datasets, forecasting scales, and model paradigms, consistently maintaining leading performance in diverse time series forecasting scenarios.

## C.3 HOW TO CAPTURE TRANSIENT DYNAMICS COMPARED TO SOTA BASELINES

This section evaluates the ability of our proposed framework, Flames, to capture transient dynamics in time series, especially short-term abrupt fluctuations. We conduct a case study on the ETTm1 dataset, as shown in Figure 7, which presents two local segments containing sharp drops and rapid recoveries. For comparison, we include predictions from two strong baselines, DTMamba and iTransformer, alongside Flames and the ground truth. Shaded regions highlight intervals with pronounced transient patterns.

The results clearly show that Flames excels at modeling these short-term dynamics. While DT-Mamba partially follows the downward trends but fails to capture the deepest troughs and fast re-

Table 6: Multivariate long-term series forecasting results on different prediction lengths $\in \{96, 192, 336, 720\}$. A lower value indicates better performance. **Bold**: best, underlined: second best.

| Dataset | | Flames | | S-Mamba | | FiLM | | FreTS | | MICN | | TimePro | | SCINet | | Autoformer | | FEDformer | |
|---|---|---|---|---|---|---|---|---|---|---|---|---|---|---|---|---|---|---|---|
| Metrics | | MSE | MAE | MSE | MAE | MSE | MAE | MSE | MAE | MSE | MAE | MSE | MAE | MSE | MAE | MSE | MAE | MSE | MAE |
| ETTh1 | 96 | 0.392 | 0.410 | 0.386 | 0.405 | 0.438 | 0.433 | 0.402 | 0.416 | 0.426 | 0.446 | **0.375** | **0.398** | 0.654 | 0.599 | 0.449 | 0.459 | 0.376 | 0.419 |
| | 192 | 0.422 | **0.428** | 0.443 | 0.437 | 0.494 | 0.466 | 0.472 | 0.462 | 0.454 | 0.464 | 0.427 | 0.429 | 0.719 | 0.631 | 0.500 | 0.482 | **0.420** | 0.448 |
| | 336 | 0.462 | **0.441** | 0.489 | 0.468 | 0.547 | 0.495 | 0.518 | 0.484 | 0.493 | 0.487 | 0.472 | 0.450 | 0.778 | 0.659 | 0.521 | 0.496 | **0.459** | 0.465 |
| | 720 | **0.452** | **0.456** | 0.502 | 0.489 | 0.586 | 0.538 | 0.573 | 0.548 | 0.526 | 0.526 | 0.476 | 0.474 | 0.836 | 0.699 | 0.514 | 0.512 | 0.506 | 0.507 |
| | Avg. | **0.432** | **0.434** | 0.455 | 0.450 | 0.516 | 0.483 | 0.491 | 0.478 | 0.475 | 0.480 | 0.438 | 0.438 | 0.747 | 0.647 | 0.496 | 0.487 | 0.440 | 0.460 |
| ETTh2 | 96 | 0.301 | 0.354 | 0.296 | 0.348 | 0.322 | 0.364 | 0.347 | 0.399 | 0.372 | 0.424 | **0.293** | **0.345** | 0.707 | 0.621 | 0.346 | 0.388 | 0.358 | 0.397 |
| | 192 | **0.365** | **0.394** | 0.376 | 0.396 | 0.405 | 0.414 | 0.480 | 0.478 | 0.492 | 0.492 | 0.367 | 0.394 | 0.860 | 0.689 | 0.456 | 0.452 | 0.429 | 0.439 |
| | 336 | **0.371** | **0.407** | 0.424 | 0.431 | 0.435 | 0.445 | 0.519 | 0.509 | 0.607 | 0.555 | 0.419 | 0.431 | 1.000 | 0.744 | 0.482 | 0.486 | 0.496 | 0.487 |
| | 720 | **0.411** | **0.430** | 0.426 | 0.444 | 0.445 | 0.457 | 0.780 | 0.638 | 0.824 | 0.655 | 0.427 | 0.445 | 1.249 | 0.838 | 0.515 | 0.511 | 0.463 | 0.474 |
| | Avg. | **0.362** | **0.396** | 0.381 | 0.405 | 0.402 | 0.420 | 0.532 | 0.606 | 0.574 | 0.531 | 0.377 | 0.403 | 0.954 | 0.723 | 0.450 | 0.459 | 0.437 | 0.449 |
| ETTm1 | 96 | **0.293** | **0.356** | 0.333 | 0.368 | 0.353 | 0.370 | 0.352 | 0.385 | 0.365 | 0.387 | 0.326 | 0.364 | 0.418 | 0.438 | 0.505 | 0.475 | 0.379 | 0.419 |
| | 192 | **0.344** | **0.378** | 0.376 | 0.390 | 0.389 | 0.387 | 0.394 | 0.406 | 0.403 | 0.408 | 0.367 | 0.383 | 0.439 | 0.450 | 0.553 | 0.496 | 0.426 | 0.441 |
| | 336 | **0.372** | **0.402** | 0.408 | 0.413 | 0.421 | 0.408 | 0.430 | 0.431 | 0.436 | 0.431 | 0.402 | 0.409 | 0.490 | 0.485 | 0.621 | 0.537 | 0.445 | 0.459 |
| | 720 | **0.439** | **0.440** | 0.475 | 0.448 | 0.481 | 0.441 | 0.494 | 0.472 | 0.489 | 0.462 | 0.469 | 0.446 | 0.595 | 0.550 | 0.671 | 0.561 | 0.543 | 0.490 |
| | Avg. | **0.362** | **0.394** | 0.398 | 0.405 | 0.412 | 0.402 | 0.418 | 0.424 | 0.423 | 0.422 | 0.391 | 0.400 | 0.485 | 0.481 | 0.588 | 0.517 | 0.448 | 0.452 |
| ETTm2 | 96 | 0.185 | 0.275 | 0.179 | 0.263 | 0.183 | 0.266 | 0.194 | 0.290 | 0.197 | 0.296 | **0.178** | **0.260** | 0.286 | 0.377 | 0.255 | 0.339 | 0.203 | 0.287 |
| | 192 | **0.239** | 0.305 | 0.250 | 0.309 | 0.248 | 0.305 | 0.283 | 0.359 | 0.284 | 0.361 | 0.242 | **0.303** | 0.399 | 0.445 | 0.281 | 0.340 | 0.269 | 0.328 |
| | 336 | **0.294** | **0.338** | 0.312 | 0.349 | 0.309 | 0.343 | 0.360 | 0.407 | 0.381 | 0.429 | 0.303 | 0.342 | 0.637 | 0.591 | 0.399 | 0.372 | 0.325 | 0.366 |
| | 720 | **0.381** | **0.392** | 0.411 | 0.406 | 0.410 | 0.400 | 0.545 | 0.516 | 0.549 | 0.522 | 0.400 | 0.399 | 0.960 | 0.735 | 0.433 | 0.432 | 0.421 | 0.415 |
| | Avg. | **0.275** | 0.328 | 0.288 | 0.332 | 0.288 | 0.328 | 0.346 | 0.393 | 0.353 | 0.402 | 0.281 | **0.326** | 0.571 | 0.537 | 0.327 | 0.371 | 0.305 | 0.349 |
| Electricity | 96 | **0.128** | **0.222** | 0.139 | 0.235 | 0.198 | 0.274 | 0.189 | 0.276 | 0.180 | 0.293 | 0.139 | 0.234 | 0.247 | 0.345 | 0.201 | 0.317 | 0.193 | 0.308 |
| | 192 | **0.151** | **0.242** | 0.159 | 0.255 | 0.198 | 0.278 | 0.191 | 0.279 | 0.189 | 0.302 | 0.156 | 0.249 | 0.257 | 0.355 | 0.222 | 0.334 | 0.201 | 0.315 |
| | 336 | **0.171** | 0.268 | 0.176 | 0.272 | 0.217 | 0.300 | 0.206 | 0.296 | 0.198 | 0.312 | 0.172 | **0.267** | 0.269 | 0.369 | 0.231 | 0.338 | 0.214 | 0.329 |
| | 720 | **0.195** | 0.293 | 0.204 | 0.298 | 0.278 | 0.356 | 0.246 | 0.332 | 0.217 | 0.330 | 0.209 | **0.299** | 0.299 | 0.390 | 0.254 | 0.361 | 0.246 | 0.355 |
| | Avg. | **0.161** | **0.256** | 0.170 | 0.265 | 0.223 | 0.302 | 0.208 | 0.296 | 0.196 | 0.309 | 0.169 | 0.262 | 0.268 | 0.365 | 0.227 | 0.338 | 0.214 | 0.327 |
| Exchange | 96 | 0.089 | **0.204** | 0.086 | 0.207 | 0.098 | 0.256 | 0.094 | 0.222 | 0.097 | 0.228 | **0.085** | **0.204** | 0.267 | 0.396 | 0.197 | 0.323 | 0.148 | 0.278 |
| | 192 | 0.175 | 0.297 | 0.182 | 0.304 | 0.282 | 0.335 | 0.222 | 0.350 | 0.288 | 0.312 | 0.178 | 0.299 | 0.351 | 0.459 | 0.300 | 0.369 | 0.271 | 0.315 |
| | 336 | 0.327 | 0.414 | 0.332 | 0.418 | 0.425 | 0.487 | 0.431 | 0.492 | 0.387 | 0.442 | 0.328 | 0.414 | 1.423 | 0.853 | 0.509 | 0.524 | 0.460 | 0.427 |
| | 720 | 0.878 | 0.696 | 0.867 | 0.703 | 0.912 | 0.732 | 1.007 | 0.764 | 0.904 | 0.716 | **0.817** | **0.679** | 1.058 | 0.797 | 1.447 | 0.941 | 1.195 | 0.695 |
| | Avg. | 0.367 | 0.403 | 0.367 | 0.408 | 0.429 | 0.453 | 0.439 | 0.457 | 0.419 | 0.425 | **0.352** | **0.399** | 0.750 | 0.626 | 0.613 | 0.539 | 0.519 | 0.429 |
| Traffic | 96 | **0.358** | **0.251** | 0.382 | 0.261 | 0.514 | 0.304 | 0.557 | 0.329 | 0.519 | 0.329 | 0.377 | 0.258 | 0.788 | 0.499 | 0.613 | 0.388 | 0.587 | 0.366 |
| | 192 | **0.382** | **0.268** | 0.396 | 0.267 | 0.528 | 0.329 | 0.569 | 0.338 | 0.537 | 0.315 | 0.398 | 0.271 | 0.789 | 0.505 | 0.616 | 0.382 | 0.604 | 0.373 |
| | 336 | **0.398** | **0.277** | 0.417 | 0.276 | 0.537 | 0.334 | 0.566 | 0.337 | 0.534 | 0.313 | 0.420 | 0.279 | 0.797 | 0.508 | 0.622 | 0.337 | 0.621 | 0.383 |
| | 720 | **0.427** | **0.296** | 0.460 | 0.300 | 0.622 | 0.363 | 0.603 | 0.357 | 0.577 | 0.325 | 0.452 | 0.297 | 0.841 | 0.523 | 0.660 | 0.408 | 0.626 | 0.382 |
| | Avg. | **0.391** | **0.273** | 0.414 | 0.276 | 0.550 | 0.333 | 0.574 | 0.340 | 0.542 | 0.316 | 0.412 | 0.276 | 0.804 | 0.509 | 0.628 | 0.379 | 0.610 | 0.376 |
| Weather | 96 | **0.152** | **0.203** | 0.165 | 0.210 | 0.195 | 0.236 | 0.183 | 0.238 | 0.198 | 0.261 | 0.166 | 0.207 | 0.221 | 0.306 | 0.266 | 0.336 | 0.217 | 0.296 |
| | 192 | **0.196** | **0.231** | 0.214 | 0.252 | 0.239 | 0.271 | 0.251 | 0.312 | 0.239 | 0.299 | 0.216 | 0.254 | 0.261 | 0.340 | 0.307 | 0.367 | 0.276 | 0.336 |
| | 336 | **0.246** | **0.285** | 0.274 | 0.297 | 0.289 | 0.306 | 0.272 | 0.316 | 0.285 | 0.316 | 0.273 | 0.296 | 0.309 | 0.378 | 0.359 | 0.395 | 0.339 | 0.380 |
| | 720 | **0.315** | **0.335** | 0.350 | 0.345 | 0.360 | 0.351 | 0.349 | 0.377 | 0.351 | 0.388 | 0.351 | 0.346 | 0.377 | 0.427 | 0.419 | 0.428 | 0.403 | 0.428 |
| | Avg. | **0.227** | **0.263** | 0.251 | 0.276 | 0.271 | 0.290 | 0.264 | 0.311 | 0.268 | 0.321 | 0.251 | 0.276 | 0.292 | 0.363 | 0.338 | 0.382 | 0.309 | 0.360 |
| $1^{st}$ Count | | 64 | | 0 | | 0 | | 0 | | 0 | | 14 | | 0 | | 0 | | 2 | |

bounds, and iTransformer produces overly smoothed predictions, Flames closely matches the ground truth, accurately reconstructing local extrema and rapid turning points.

The Laplace Transform plays a key role in this improvement by emphasizing high-frequency and rapidly changing components. An ablation study removing the inverse Laplace transform (Flames(-ILT)) further confirms its importance: without ILT, predictions become delayed and distorted in fluctuation-heavy regions.

Overall, these findings demonstrate that Flames provides superior modeling of transient and highly dynamic patterns that baseline models struggle to capture.

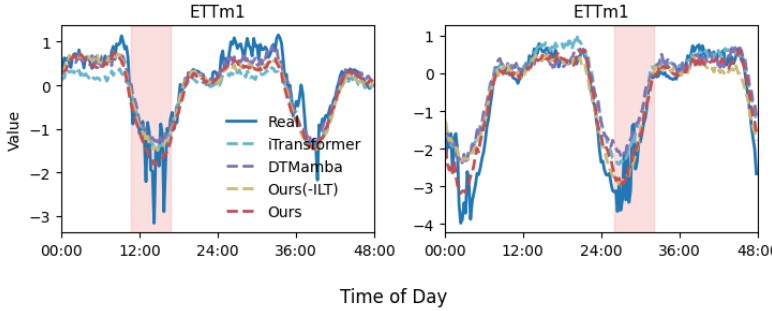

Figure 6: Case study on Flames for modeling transient dynamics like short-term fluctuations

Table 7: More Baseliens, including three Mamba-based models, one frequency-filter model, transformer-based, and linear-based models.

| Dataset | | Flames | | FLDmamba | | SiMBA | | Time-SSM | | FilterNet | | Crossformer | | Rlinear | |
|---|---|---|---|---|---|---|---|---|---|---|---|---|---|---|---|
| Metrics | | MSE | MAE | MSE | MAE | MSE | MAE | MSE | MAE | MSE | MAE | MSE | MAE | MSE | MAE |
| ETTh1 | 96 | 0.392 | 0.410 | **0.374** | 0.393 | 0.379 | 0.395 | 0.377 | 0.394 | 0.378 | **0.389** | 0.423 | 0.448 | 0.386 | 0.395 |
| | 192 | **0.422** | 0.428 | 0.427 | **0.422** | 0.432 | 0.424 | 0.423 | 0.424 | 0.442 | 0.423 | 0.471 | 0.474 | 0.437 | 0.424 |
| | 336 | 0.462 | 0.441 | **0.447** | 0.441 | 0.473 | 0.443 | 0.466 | **0.437** | 0.490 | 0.446 | 0.570 | 0.546 | 0.479 | 0.446 |
| | 720 | **0.452** | 0.456 | 0.469 | 0.463 | 0.483 | 0.469 | 0.452 | **0.448** | 0.492 | 0.463 | 0.653 | 0.621 | 0.481 | 0.470 |
| | Avg. | 0.432 | 0.434 | 0.434 | 0.430 | 0.442 | 0.433 | **0.430** | **0.426** | 0.451 | 0.430 | 0.529 | 0.522 | 0.446 | 0.434 |
| ETTh2 | 96 | 0.301 | 0.354 | 0.287 | 0.337 | 0.290 | 0.339 | 0.290 | 0.341 | **0.280** | **0.328** | 0.745 | 0.584 | 0.288 | 0.338 |
| | 192 | **0.365** | 0.394 | 0.370 | 0.388 | 0.373 | 0.390 | 0.368 | **0.387** | 0.369 | 0.395 | 0.877 | 0.656 | 0.374 | 0.390 |
| | 336 | **0.371** | 0.407 | 0.412 | 0.425 | 0.376 | **0.406** | 0.416 | 0.430 | 0.412 | 0.421 | 1.043 | 0.731 | 0.415 | 0.426 |
| | 720 | **0.411** | **0.430** | 0.419 | 0.438 | 0.417 | 0.431 | 0.424 | 0.439 | 0.422 | 0.437 | 1.104 | 0.763 | 0.420 | 0.440 |
| | Avg. | **0.362** | **0.396** | 0.372 | 0.396 | 0.364 | 0.392 | 0.375 | 0.399 | 0.371 | 0.395 | 0.942 | 0.684 | 0.374 | 0.399 |
| ETTm1 | 96 | **0.293** | **0.356** | 0.318 | 0.360 | 0.324 | 0.360 | 0.329 | 0.365 | 0.313 | 0.358 | 0.404 | 0.426 | 0.355 | 0.376 |
| | 192 | **0.344** | **0.378** | 0.365 | 0.384 | 0.363 | 0.382 | 0.370 | 0.379 | 0.369 | 0.383 | 0.450 | 0.451 | 0.391 | 0.392 |
| | 336 | **0.372** | **0.402** | 0.404 | 0.409 | 0.395 | 0.405 | 0.396 | 0.402 | 0.399 | 0.406 | 0.532 | 0.515 | 0.424 | 0.415 |
| | 720 | **0.439** | 0.440 | 0.464 | 0.441 | 0.451 | **0.437** | 0.449 | 0.440 | 0.466 | 0.444 | 0.666 | 0.589 | 0.487 | 0.450 |
| | Avg. | **0.362** | **0.394** | 0.389 | 0.399 | 0.383 | 0.396 | 0.386 | 0.396 | 0.387 | 0.398 | 0.513 | 0.495 | 0.414 | 0.408 |
| ETTm2 | 96 | 0.185 | 0.275 | **0.173** | **0.253** | 0.177 | 0.263 | 0.176 | 0.260 | 0.171 | 0.257 | 0.287 | 0.366 | 0.182 | 0.265 |
| | 192 | **0.239** | 0.305 | 0.240 | **0.299** | 0.245 | 0.306 | 0.240 | 0.305 | 0.240 | 0.300 | 0.414 | 0.492 | 0.246 | 0.304 |
| | 336 | **0.294** | 0.338 | 0.301 | **0.307** | 0.304 | 0.343 | 0.305 | 0.344 | 0.297 | 0.339 | 0.597 | 0.542 | 0.307 | 0.342 |
| | 720 | **0.381** | **0.392** | 0.401 | 0.397 | 0.400 | 0.399 | 0.406 | 0.405 | 0.392 | 0.393 | 1.730 | 1.042 | 0.407 | 0.398 |
| | Avg. | **0.275** | 0.328 | 0.279 | **0.314** | 0.282 | 0.328 | 0.283 | 0.328 | 0.275 | 0.322 | 0.757 | 0.611 | 0.286 | 0.327 |
| Electricity | 96 | **0.128** | **0.222** | 0.137 | 0.234 | 0.165 | 0.253 | \ | \ | 0.183 | 0.259 | 0.219 | 0.314 | 0.201 | 0.281 |
| | 192 | **0.151** | **0.242** | 0.158 | 0.251 | 0.173 | 0.262 | \ | \ | 0.189 | 0.267 | 0.231 | 0.322 | 0.201 | 0.283 |
| | 336 | **0.171** | **0.268** | 0.182 | 0.273 | 0.188 | 0.277 | \ | \ | 0.205 | 0.284 | 0.246 | 0.337 | 0.215 | 0.298 |
| | 720 | **0.195** | **0.293** | 0.200 | 0.292 | 0.214 | 0.305 | \ | \ | 0.246 | 0.317 | 0.280 | 0.363 | 0.257 | 0.331 |
| | Avg. | **0.161** | **0.256** | 0.170 | 0.263 | 0.185 | 0.274 | \ | \ | 0.206 | 0.282 | 0.244 | 0.334 | 0.219 | 0.298 |
| Exchange | 96 | 0.089 | 0.204 | 0.085 | 0.205 | 0.091 | 0.211 | **0.083** | **0.202** | 0.087 | 0.206 | 0.256 | 0.367 | 0.093 | 0.217 |
| | 192 | 0.175 | 0.297 | 0.175 | 0.297 | 0.186 | 0.310 | **0.170** | **0.295** | 0.236 | 0.352 | 0.470 | 0.509 | 0.184 | 0.307 |
| | 336 | **0.327** | **0.414** | 0.337 | 0.418 | 0.352 | 0.422 | 0.334 | 0.418 | 0.384 | 0.455 | 1.268 | 0.883 | 0.351 | 0.432 |
| | 720 | 0.878 | 0.696 | 0.825 | 0.683 | 0.867 | 0.702 | **0.824** | **0.677** | 0.933 | 0.736 | 1.767 | 1.068 | 0.886 | 0.714 |
| | Avg. | 0.367 | 0.403 | **0.351** | 0.400 | 0.374 | 0.411 | 0.352 | **0.398** | 0.410 | 0.437 | 0.940 | 0.707 | 0.379 | 0.418 |
| Traffic | 96 | **0.358** | **0.251** | 0.395 | 0.279 | 0.468 | 0.268 | \ | \ | 0.433 | 0.278 | 0.644 | 0.429 | 0.649 | 0.389 |
| | 192 | **0.382** | **0.268** | 0.401 | 0.278 | 0.413 | 0.317 | \ | \ | 0.448 | 0.283 | 0.665 | 0.431 | 0.601 | 0.366 |
| | 336 | **0.398** | **0.277** | 0.410 | 0.283 | 0.529 | 0.284 | \ | \ | 0.463 | 0.289 | 0.674 | 0.420 | 0.609 | 0.369 |
| | 720 | **0.427** | **0.296** | 0.447 | 0.303 | 0.564 | 0.297 | \ | \ | 0.492 | 0.305 | 0.683 | 0.424 | 0.647 | 0.387 |
| | Avg. | **0.391** | **0.273** | 0.413 | 0.286 | 0.494 | 0.292 | \ | \ | 0.459 | 0.289 | 0.667 | 0.426 | 0.627 | 0.378 |
| Weather | 96 | **0.152** | **0.203** | 0.152 | 0.203 | 0.176 | 0.219 | 0.167 | 0.212 | 0.155 | 0.193 | 0.195 | 0.271 | 0.192 | 0.232 |
| | 192 | **0.196** | **0.231** | 0.203 | 0.250 | 0.222 | 0.260 | 0.217 | 0.255 | 0.204 | 0.241 | 0.209 | 0.277 | 0.240 | 0.271 |
| | 336 | **0.246** | **0.285** | 0.242 | 0.285 | 0.275 | 0.297 | 0.274 | 0.294 | 0.265 | 0.285 | 0.273 | 0.332 | 0.292 | 0.307 |
| | 720 | **0.315** | **0.335** | 0.317 | 0.336 | 0.350 | 0.349 | 0.351 | 0.345 | 0.354 | 0.342 | 0.379 | 0.401 | 0.364 | 0.353 |
| | Avg. | **0.227** | **0.263** | 0.229 | 0.269 | 0.256 | 0.281 | 0.252 | 0.276 | 0.245 | 0.265 | 0.264 | 0.320 | 0.272 | 0.291 |
| 1st Count | | **54** | | 9 | | 2 | | 12 | | 3 | | 0 | | 0 | |

Table 8: Sensitivity experiments of mask_ratio.

| mask_ratio | p=0.01 | p=0.1 | p=0.2 | p=0.25 | p=0.3 | p=0.5 |
|---|---|---|---|---|---|---|
| Weather (avg.) | 0.238 | 0.233 | 0.229 | **0.227** | 0.231 | 0.234 |
| ETTm1 (avg.) | 0.377 | 0.372 | 0.365 | **0.362** | 0.366 | 0.369 |

## C.4 SENSITIVITY ANALYSIS

Flames involves several tunable hyperparameters. A properly chosen mask_ratio not only facilitates effective representation learning but also enhances robustness to data instability and missing values in reality. To identify the optimal setting, we performed a sensitivity analysis on mask_ratio on the ETTh1 and Weather datasets. Table 8 shows that the average MSE is lowest when mask_ratio = 0.25, indicating the best overall performance.

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
