# OpenReview forum: "Flames: Multi-Scale Mamba with Adaptive Fourier Filters and Laplace Transform for Time Series Forecasting"
_ICLR.cc/2026/Conference — Submitted to ICLR 2026_

### Official Review · Reviewer_nUNg · 2025-10-31

**Soundness:** 2
**Presentation:** 2
**Contribution:** 1
**Rating:** 2
**Confidence:** 4

**Summary:**

This paper proposes a novel multi-scale Mamba framework that incorporates adaptive Fourier filters to cover all frequencies at each scale and a Laplace-transform module to capture short-term fluctuations.

**Strengths:**

1. This paper proposes an Adaptive Fourier Filter Module (AFFM) encoder, which employs adaptive Fourier filtering to capture multi-scale periodic patterns and to reduce noise.
2. This paper applies the inverse Laplace transform to extract short-term dynamics.

**Weaknesses:**

1. The paper is poorly organized and lacks clarity in writing, making it difficult to follow. In addition, multiple inconsistencies in symbol usage should be carefully reviewed, for example, the one observed on Line 232.
2. The motivation of the paper is not clear. In the Introduction, the authors argue that using Mamba as the backbone faces three key challenges: (i) multiscale periodicity, (ii) data noise, and (iii) transient dynamics. However, no strong prior studies or preliminary experiments are provided to substantiate these claims. Other important assertions, such as “Linear struggles with noisy data and fails to capture long-term dependencies effectively” are likewise unsupported. In addition, the manuscript does not offer a substantive discussion of whether alternative backbones (e.g., Transformer, MLP) can address these three challenges, nor does it provide comparative analysis to justify the choice of Mamba.
3. It would strengthen the work to include additional frequency filter-based baselines, such as FilterNet [1], TSLANet [2], which are closely related to the proposed approach.

[1] FilterNet: Harnessing Frequency Filters for Time Series Forecasting. NeurIPS, 2024.
[2] TSLANet: Rethinking Transformers for Time Series Representation Learning. ICML, 2024.

**Questions:**

pls refer to weakness.

---

> ### Author Response · Authors · 2025-11-21
> **Response to reviewer nUNg (1/2)**
>
> Thank you for taking the time and providing thoughtful and valuable feedback. Below, we address each of your comments and questions in detail.
>
> **Q1: Inconsistencies in symbol usage**
>
> R1: We carefully re-read the manuscript and have made a set of systematic revisions to improve readability and unify the symbols.
> - To understand the whole algorithm process conveniently, we supplemented the combination of these three modules (Mamba, Fourier filter, and Laplace transform) in Section B in Appendix.
> - The 232 line symbols are explained in detail.
>
> (1) For your convenience, here is Section B in Appendix:
>
> This section provides a combined view that clarifies how jointly define a flexible yet structured forecasting operator.
>
> **Multi-scale decomposition.** Given an input time series $x \in \mathbb{R}^{T \times C}$, Flames constructs multi-scale inputs $\{x_m\}_{m=0}^{M}$ via the recursive downsampling
>
> $$
>     x_m = {Conv}(x_{m-1}, \mathrm{stride}=2), \quad m \in {1,\dots,M},
> $$
>
> where $x_0$ = x and $x_m \in \mathbb{R}^{\lfloor T / 2^m \rfloor \times C}$. An embedding layer then projects each scale into deep features:
>
> $$
> X' = \mathrm{Embed}(X) = \{x'_0,\dots,x'_M\},\quad
> x'_m \in \mathbb{R}^{\lfloor T / 2^m \rfloor \times d_{\text{model}}}.
> $$
>
>
> which serve as the inputs of AFFM at different resolutions.
>
> **AFFM.** For a given scale m, let $x'_m$ denote the embedded sequence. AFFM first transforms $x'_m$ from the time domain to the frequency domain by applying a 1D FFT:
>
> $$
>     X_{F,m} = \mathcal{F}[x'_m],
> $$
> where $\mathcal{F}[\cdot]$ is the FFT operator.
>
> As detailed in Sec.3.2, adaptive high-frequency denoising and learnable global/local filters are then applied in the frequency domain:
>
> $$
>     X_{F,m}^{\mathrm{high}} = X_{F,m} \odot (P > \theta_{\mathrm{high}}),
> $$
>
> The global filter:
>
> $$
>     X_{G,m} = \mathcal{M}(X_{F,m})_{G} \odot X_{F,m},
> $$
>
> The local filter:
>
> $$
>     X_{L,m}^{\mathrm{high}} = \mathcal{M}(X_{F,m}^{\mathrm{high}})_{L} \odot X_{F,m}^{\mathrm{high}},
> $$
>
> and the filtered spectrum at scale $m$ is obtained by
>
> $$
>     X_{mixed,m} = X_{G,m} + X_{L,m}^{\mathrm{high}}.
> $$
>
> Finally, an inverse FFT (IFFT) maps the mixed spectrum back to the time domain,
>
> $$
>     X_m^{T} = \mathcal{F}^{-1}[X_{mixed,m}],
> $$
>
> where $\mathcal{F}^{-1}[\cdot]$ denotes IFFT and $X^{T}_m$ has the same temporal shape as $x'_m$.
> By the convolution theorem, the element-wise multiplication in the frequency domain is equivalent to a convolution in the time domain, i.e., there exists a data-dependent kernel $h_m$ such that
>
> $$
>     X^{T}_m(t) = (h_m * x'_m)(t),
> $$
>
> where $*$ denotes convolution along the temporal dimension.
>
> Therefore, AFFM can be interpreted as learning a \emph{spectrally shaped kernel} $h_m$ that filters out noise and emphasizes task-relevant frequency bands for each scale, rather than only applying a point-wise nonlinearity on top of Mamba states.
>
> **Inverse Laplace Transform (ILT).** On top of the AFFM-filtered representation, Flames employs the inverse Laplace transform (ILT) head discussed in Sec.~3.4. Eq.13 parameterizes the forecast $u(t)$ as a superposition of damped sinusoidal modes,
>
> $$
>     u(t) = \sum_{n=1}^{M} A_n e^{-\sigma_n t} \cos(\omega_n t + \phi_n),
> $$
>
> where $A_n$, $\sigma_n$, $\omega_n$, and $\phi_n$ are learnable functions of the historical sequence.
>
> For each scale $m$, we instantiate this parameterization with its own parameters
> \(
> \{A_{m,n}, \sigma_{m,n}, \omega_{m,n}, \phi_{m,n}\}_{n=1}^{M}
> \)
> that are predicted from $X^{T}_m$:
>
> $$
>     u^{(m)}(t) = \sum_{n=1}^{M} A_{m,n} e^{-\sigma_{m,n} t} \cos(\omega_{m,n} t + \phi_{m,n}),
> $$
>
> so that small $\sigma_{m,n}$ encode long-term periodic trends and large $\sigma_{m,n}$ capture short-lived transient dynamics at scale $m$.
>
> **Overall operator.** Combining the two views above, the contribution of scale $m$ to the prediction can be written as
>
> $$
>     u^{(m)}(t) = \sum_{n=1}^{M} A_{m,n} \big( h_m * x'_m \big)(t)
>       e^{-\sigma_{m,n} t} \cos(\omega_{m,n} t + \phi_{m,n}),
> $$
>
> where $h_m$ is the AFFM-induced kernel for scale $m$ and $x'_m$ is the embedded multi-scale input. The outputs $\{u^{(m)}\}_{m=0}^{M}$ from all scales are then fused by the multi-scale fusion module (Sec.~3.3), which concatenates the scale-wise representations and applies a linear projection to obtain the final forecast.
>
> (2) Here is the symbol explanation of Line 230 and Line 232：
> In our formulation, X is consistently used for multivariate sequence representations, ``X_{MSF}'' denotes the multi-scale fusion output (Line 230), $M$ denotes the number of time scales, which is correct in our manuscript (Line 232):
>
> $$
>     X_{\text{MSF}} = \mathrm{Linear}\big(\mathrm{concat}(u_0, u_1, \dots, u_M)\big).
> $$
>
> Overall, we highly value the readability of the paper, and we appreciate the reviewer’s feedback, which motivated us to improve the structure and notation of the manuscript. We hope this could address these concerns.

---

> ### Author Response · Authors · 2025-11-23
> **Response to reviewer nUNg (2/2)**
>
> **Q2: Motivation**
>
> R2: In the Introduction, we already discuss the challenges of multi-scale periodicity, noise, and transient dynamics in TSF. In Section 4.3 we provide ablation studies showing the contribution of the three corresponding modules. In Section 4.4, we further inject Gaussian noise with different intensities into Weather and ETTh1 and compare Transformer, PatchTST, and our method. The results show that noise significantly degrades existing models, while Flames remains more robust. Building on this, we have enhanced the Introduction and Related Work in the revised manuscript to clarify the motivation of the three challenges. Specifically:
>
> (1) Motivation
>
> - We add more prior references to support the three challenges in the Introduction, and “More Motivation”, which includes multiscale modeling, noise, and transient dynamics in Section B.2 in Appendix.
>
> (2) Linear models under noise and long-term dependencies
>
> In the revised version, we back up this statement with appropriate prior work, and we explicitly point out in Table 1, we compare Flames with DLinear, RLinear, TimeMixer, and other linear models on 8 datasets and 4 forecasting horizons. On high-dimensional or noisy datasets (e.g., Electricity, Traffic, Weather) and for long horizons (336/720), DLinear and RLinear consistently lag behind Flames and several Transformer/Mamba baselines, which empirically confirms the limitations of purely linear mappings under noise and long-term dependencies.
>
> (3) Alternative backbones (Transformer / MLP) and the justification for choosing Mamba on theory and experiments: In the Introduction and Related Work, we have provided a summary and comparison of the advantages and limitations of Transformer-, MLP-, and Mamba-based backbones. In Section 4, we already compared Flames against SOTA baselines built on Transformers, MLPs, and other Mamba variants, thus covering diverse backbone designs. The results show that Flames achieves the best or second-best performance in most scenarios, while maintaining a relatively small number of parameters and low computational cost.
>
> Why we choose Mamba as the backbone: In the original manuscript, we do not claim Transformers or MLPs cannot address these challenges. Our choice of Mamba is based on the following considerations: (i) Mamba offers linear-time complexity and good scalability for long sequence modeling; (ii) existing Mamba variants pay relatively little attention to frequency-domain modeling and transient dynamics, leaving considerable room for improvement; and (iii) our proposed combination of multi-scale design, adaptive Fourier filtering, and Laplace-based operator naturally complements the selective state-space structure of Mamba, yielding a lightweight yet effective alternative, which will be added in Section B.3 in Appendix in the revised version.
>
>
> **Q3: More frequency filter-based baselines**
>
> R3: Our Flames, FilterNet, and TSLANet all employ frequency-domain filters for denoising. To fairly compare their performance, we evaluate them on three higher-dimensional and more complex datasets: Electricity (321 dim),Traffic (862 dim), and Weather (21 dim). Overall, Flames clearly outperforms the other two models, achieving the best results in 18 out of 30 forecasting settings. This indicates that Flames has better scalability and robustness on real-world multivariate time series with high dimensionality and strong noise, and can more effectively capture both temporal dependencies and cross-variable interactions. Please see Section C.2 in the Appendix.
>
> |Dataset|Metrics|Flames| |FilterNet| |TSLANet| |
> |---|---|---|---|---|---|---|---|
> | | |MSE|MAE|MSE|MAE|MSE|MAE|
> |Electricity|96|**0.128**|**0.222**|0.183|0.259|0.136|0.229|
> ||192|**0.151**|**0.242**|0.189|0.267|0.152|0.244|
> ||336|0.171|0.268|0.205|0.284|**0.168**|**0.262**|
> ||720|**0.195**|**0.293**|0.246|0.317|0.205|0.293|
> ||Avg.|**0.161**|**0.256**|0.206|0.282|0.165|0.257|
> |Traffic|96|**0.358**|**0.251**|0.433|0.278|0.372|0.261|
> ||192|**0.382**|0.268|0.448|0.283|0.388|**0.266**|
> ||336|0.398|0.277|0.463|0.289|**0.394**|**0.269**|
> ||720|**0.427**|0.296|0.492|0.305|0.430|**0.289**|
> ||Avg.|**0.391**|0.273|0.459|0.289|0.396|**0.271**|
> |Weather|96|0.152|0.203|0.155|0.193|**0.148**|**0.197**|
> ||192|0.196|**0.231**|0.204|0.241|**0.193**|0.241|
> ||336|0.246|0.285|0.265|0.285|**0.245**|**0.282**|
> ||720|**0.315**|**0.335**|0.354|0.342|0.325|0.337|
> ||Avg.|**0.227**|**0.263**|0.245|0.265|0.228|0.264|
> |$1^{st}$ Count||18||0||12||

---

> ### Author Response · Authors · 2025-11-28
> **Response to reviewer nUNg**
>
> Dear Reviewer,
>
> Thanks a lot reviewer for your valuable time to read this response. We believe that the additional information we provided in our rebuttal—such as new experimental results, further details, and clarifications on misunderstandings—addresses your key questions. Please let us know if our response has adequately addressed your concerns. We are more than willing to discuss any points that may still be unclear. We hope that the improvements and clarifications provided in our response will positively influence your assessment of our work.
>
> Best, Authors of Paper 4431

---

### Official Review · Reviewer_Q1SQ · 2025-10-31

**Soundness:** 2
**Presentation:** 3
**Contribution:** 2
**Rating:** 6
**Confidence:** 2

**Summary:**

This paper proposed a framework for long-term time series forecasting. The model enhances the Mamba state space model by integrating three components. Experimental results on eight benchmark datasets show that Flames achieves good performance, outperforming Transformer- and MLP-based baselines with better noise robustness, scalability, and computational efficiency.

**Strengths:**

- Combines Fourier and Laplace transforms with state space modeling for a unique approach to capturing multi-scale and transient temporal patterns.
- Demonstrates consistent improvement over leading baselines across multiple datasets and noise levels, with extensive ablation and robustness studies.
- Achieves linear computational complexity, lower parameter counts, and faster inference while maintaining high accuracy.

**Weaknesses:**

- The combination of multi-scale processing, Mamba, Adaptive Fourier Filters, and the Inverse Laplace Transform makes the overall model architecture intricate and potentially complex to implement and fine-tune compared to simpler linear or pure Mamba models The formulation, particularly for the Laplace transform integration, is dense.
- While claims about interpretability are made, there is little empirical validation of how the model captures transient dynamics or multi-scale features in practice.
- The paper notes that the optimal number of scales varies with the prediction length, suggesting that a fixed choice (like M=3 for efficiency) may not be universally optimal and would require a trade-off analysis for new datasets or forecasting horizons.

**Questions:**

- How to simplify or modularize the integration of multi-scale processing, Adaptive Fourier Filters, and the Inverse Laplace Transform to make the model easier to implement and fine-tune?
- What experimental approaches or visual analyses could be added to empirically validate how the model captures transient dynamics and multi-scale temporal features?
- How can the model adaptively determine or learn the optimal number of scales for different datasets or forecasting horizons wthout manual tuning?

---

> ### Author Response · Authors · 2025-11-26
> **Response to reviewer Q1SQ**
>
> Thank you for taking the time and providing thoughtful and valuable feedback. Below, we address each of your comments and questions in detail.
>
> **Q1: How to simplify or modularize the integration of multi-scale processing, Adaptive Fourier Filters, and the Inverse Laplace Transform to make the model easier to implement and fine-tune?**
>
> R1: While Flames combines multi-scale processing, adaptive Fourier filters, and an inverse Laplace head, in implementation these are three lightweight, modular components that are added on top of a standard Mamba backbone:
>
> (1) Multi-scale block as a pluggable front-end. The multi-scale pathway is implemented as a stack of strided 1D convolutions that produces downsampled versions of the input, followed by a simple fusion layer. This block can be turned on/off with a single flag (e.g., use_multiscale=True/False), leaving the core Mamba implementation unchanged.
>
> (2)Adaptive Fourier Filter (AFF) as a light-weight wrapper. In a standard Mamba block, the discretization step $\Delta$ is computed from the input $x$ by a small network $f(\cdot)$, and then fed into the usual SSM update
>
> $$
> s_{t+1} = A_{\Delta} s_t + B_{\Delta} u_t,\quad y_t = C_{\Delta} s_t.
> $$
>
> In Flames we keep these SSM equations exactly the same and only post-process the step size $\Delta$ with a light-weight filter. Concretely, we first compute
>
> $$
> \Delta = f(x) \in \mathbb{R}^{B \times T \times d},
> $$
>
> then pass it through AFF to obtain a filtered version $\Delta_F = \text{AFF}(\Delta)$, and use $\Delta_F$ in place of $\Delta$ in the SSM update.
>
> AFF itself is an independent module that applies a 1D FFT along the temporal dimension of $\Delta$, multiplies the spectrum by a learnable frequency-domain mask (implemented by two tiny $1 \times 1$ convolutions for global and local filtering), and then applies an inverse FFT to map back to the time domain. From an implementation perspective, this amounts to changing a single line of code from delta = f(x) to delta = aff(f(x)), while the rest of the Mamba/SSM implementation is left untouched. In the revised version, we will add concise pseudocode in the appendix showing how to wrap an existing Mamba cell in this way so that practitioners can drop in AFF with a one-line modification. We added Pseudocode for Mamba with Adaptive Fourier Filter (AFF) in Section B.4 in the Appendix.
>
>
>
> (3) Inverse Laplace head as an optional output layer. The inverse Laplace module is used only as a prediction head: given the final representation, it predicts a small set of mode parameters and evaluates a closed-form sum of decaying sinusoids. It does not modify the backbone or training loop, and can be swapped out for a standard linear head if desired.
>
> For easier understanding, we introduced the combination of these three modules in the revised version. Please see Section B.1 in the appendix.
>
>
> **Q2: What experimental approaches or visual analyses could be added to empirically validate how the model captures transient dynamics and multi-scale temporal features?**
> R2: We conducted a case-study experiment to illustrate how our method addresses the challenges of transient dynamics, as shown in Figure 6 (Section C.3 in the Appendix) of the revised version. The comparison with DTMamba confirms that our model can more accurately capture transient behaviors. Moreover, when compared against both DTMamba and iTransformer, our method exhibits more robust performance. We further evaluate Flames by removing the inverse Laplace transform (ILT), as also shown in Figure 6. The resulting degradation in modeling short-term fluctuations confirms the critical role of ILT in handling transient dynamics.
>
> **Q3: How can the model adaptively determine or learn the optimal number of scales for different datasets or forecasting horizons without manual tuning?**
>
> R3: In our current submission, the number of scales M is selected by manual tuning, and we show in a sensitivity analysis that performance is relatively stable for M∈{1,2,3,4,5} on ETTm1, with M=3 providing a good trade-off between accuracy and efficiency across datasets. We agree that further reducing manual tuning is desirable, we can use the grid search method, or the well-known heuristic algorithm, such as genetic algorithm, particle swarm optimization algorithm, etc. However, heuristic algorithms tend to greatly increase the amount of computation and computational load.

---

### Official Review · Reviewer_A1tm · 2025-11-01

**Soundness:** 3
**Presentation:** 3
**Contribution:** 2
**Rating:** 4
**Confidence:** 4

**Summary:**

The paper proposes FLAMES, a long-term time-series forecasting model. The FLAMES augments Mamba with three components. (i) multi-scale feature extraction, (ii) Adaptive Fourier Filter Mamba that performs FFT-domain masking and global/local learnable frequency mixing to denoise and enhance periodic structure, and (iii) Laplace transform to capture transient dynamics. Experiments on multiple benchmarks report consistent gains over baselines. The paper also conducts ablations for each component, robustness to synthetic noise, and a scalability analysis to show the superiority of FLAMES.

**Strengths:**

+ The time series forecasting, particularly long-term time series forecasting is an important and practical problem.

+ The evaluation is comprehensive. The experimental results comprehensively cover multiple datasets and horizons with ablations, look-back sensitivity, robustness to noise, and scalability analysis.

**Weaknesses:**

- The method is a careful combination of known tools (such as Mamba, Fourier filter, and Laplace transform) and rather than a fundamentally new learning principle. The paper would benefit from a more formal analysis combing these modules together.

- The paper does not compare many Mamba-based time series forecaster despite mainly built on Mamba backbone. The only relevant baseline is DTMamba. It is not clear why the authors ignore many other Mamba-based time series forecasting papers, even if they are earlier or have been accepted by decent conferences. Just list some examples as below:

[1] Ahamed et al. TimeMachine: A Time Series is Worth 4 Mambas for Long-term Forecasting. ECAI. 2024

[2] Xu et al. SST: Multi-Scale Hybrid Mamba-Transformer Experts for Long-Short Range Time Series Forecasting. CIKM. 2025

[3] Patro et al. SiMBA: Simplified Mamba-Based Architecture for Vision and Multivariate Time series. arxiv. 2024.

[4] Hu et al. Time-SSM: Simplifying and Unifying State Space Models for Time Series Forecasting. arxiv, 2024.

**Questions:**

See the weaknesses.

---

> ### Author Response · Authors · 2025-11-20
> **Response to reviewer A1tm (1/2)**
>
> Thank you for taking the time and providing thoughtful and valuable feedback.
>
> **Q1: Combined modules of Flames together.**
>
> R1: After discussing in detail, we agree with the reviewer, and we have added a section in the appendix combining these modules of Flames (Section B.1 of appendix).
>
> For your convenience, here is the more analysis combining the modules together paragraph we have added:
>
> This section provides a combined view that clarifies how jointly define a flexible yet structured forecasting operator.
>
> **Multi-scale decomposition.** Given an input time series $x \in \mathbb{R}^{T \times C}$, Flames constructs multi-scale inputs $\{x_m\}_{m=0}^{M}$ via the recursive downsampling
>
> $$
>     x_m = {Conv}(x_{m-1}, \mathrm{stride}=2), \quad m \in {1,\dots,M},
> $$
>
> where $x_0$ = x and $x_m \in \mathbb{R}^{\lfloor T / 2^m \rfloor \times C}$. An embedding layer then projects each scale into deep features:
>
> $$
> X' = \mathrm{Embed}(X) = \{x'_0,\dots,x'_M\},\quad
> x'_m \in \mathbb{R}^{\lfloor T / 2^m \rfloor \times d_{\text{model}}}.
> $$
>
>
> which serve as the inputs of AFFM at different resolutions.
>
> **AFFM.** For a given scale m, let $x'_m$ denote the embedded sequence. AFFM first transforms $x'_m$ from the time domain to the frequency domain by applying a 1D FFT:
>
> $$
>     X_{F,m} = \mathcal{F}[x'_m],
> $$
> where $\mathcal{F}[\cdot]$ is the FFT operator.
>
> As detailed in Sec.3.2, adaptive high-frequency denoising and learnable global/local filters are then applied in the frequency domain:
>
> $$
>     X_{F,m}^{\mathrm{high}} = X_{F,m} \odot (P > \theta_{\mathrm{high}}),
> $$
>
> The global filter:
>
> $$
>     X_{G,m} = \mathcal{M}(X_{F,m})_{G} \odot X_{F,m},
> $$
>
> The local filter:
>
> $$
>     X_{L,m}^{\mathrm{high}} = \mathcal{M}(X_{F,m}^{\mathrm{high}})_{L} \odot X_{F,m}^{\mathrm{high}},
> $$
>
> and the filtered spectrum at scale $m$ is obtained by
>
> $$
>     X_{mixed,m} = X_{G,m} + X_{L,m}^{\mathrm{high}}.
> $$
>
> Finally, an inverse FFT (IFFT) maps the mixed spectrum back to the time domain,
>
> $$
>     X_m^{T} = \mathcal{F}^{-1}[X_{mixed,m}],
> $$
>
> where $\mathcal{F}^{-1}[\cdot]$ denotes IFFT and $X^{T}_m$ has the same temporal shape as $x'_m$.
> By the convolution theorem, the element-wise multiplication in the frequency domain is equivalent to a convolution in the time domain, i.e., there exists a data-dependent kernel $h_m$ such that
>
> $$
>     X^{T}_m(t) = (h_m * x'_m)(t),
> $$
>
> where $*$ denotes convolution along the temporal dimension.
>
> Therefore, AFFM can be interpreted as learning a \emph{spectrally shaped kernel} $h_m$ that filters out noise and emphasizes task-relevant frequency bands for each scale, rather than only applying a point-wise nonlinearity on top of Mamba states.
>
> **Inverse Laplace Transform (ILT).** On top of the AFFM-filtered representation, Flames employs the inverse Laplace transform (ILT) head discussed in Sec.~3.4. Eq.13 parameterizes the forecast $u(t)$ as a superposition of damped sinusoidal modes,
>
> $$
>     u(t) = \sum_{n=1}^{M} A_n e^{-\sigma_n t} \cos(\omega_n t + \phi_n),
> $$
>
> where $A_n$, $\sigma_n$, $\omega_n$, and $\phi_n$ are learnable functions of the historical sequence.
>
> For each scale $m$, we instantiate this parameterization with its own parameters
> \(
> \{A_{m,n}, \sigma_{m,n}, \omega_{m,n}, \phi_{m,n}\}_{n=1}^{M}
> \)
> that are predicted from $X^{T}_m$:
>
> $$
>     u^{(m)}(t) = \sum_{n=1}^{M} A_{m,n} e^{-\sigma_{m,n} t} \cos(\omega_{m,n} t + \phi_{m,n}),
> $$
>
> so that small $\sigma_{m,n}$ encode long-term periodic trends and large $\sigma_{m,n}$ capture short-lived transient dynamics at scale $m$.
>
> **Overall operator.** Combining the two views above, the contribution of scale $m$ to the prediction can be written as
>
> $$
>     u^{(m)}(t) = \sum_{n=1}^{M} A_{m,n} \big( h_m * x'_m \big)(t)
>       e^{-\sigma_{m,n} t} \cos(\omega_{m,n} t + \phi_{m,n}),
> $$
>
> where $h_m$ is the AFFM-induced kernel for scale $m$ and $x'_m$ is the embedded multi-scale input. The outputs $\{u^{(m)}\}_{m=0}^{M}$ from all scales are then fused by the multi-scale fusion module (Sec.~3.3), which concatenates the scale-wise representations and applies a linear projection to obtain the final forecast.

---

> ### Author Response · Authors · 2025-11-21
> **Response to reviewer A1tm (2/2)**
>
> **Q2: More Mamba-based baselines**
>
> R2: Since Flames is a multi-scale model, most of our baselines are multi-scale architectures. We also include other Mamba-based methods (e.g., S-Mamba) in Table 6 of the appendix and will add more Mamba-based baselines in the revised version. Following the reviewer’s suggestion, we replaced Crossformer and RLinear with the published TimeMachine and SST in Table 1, and moved Crossformer and RLinear to Section C.2 in the Appendix. The corresponding results are shown below:
>
> |Dataset|Horizon|Flames||${FLDmamba}^{[1]}$||TimeMachine||SST||SiMBA||Time-SSM||
> |-|-|-|-|-|-|-|-|-|-|-|-|-|-|
> || |MSE|MAE|MSE|MAE|MSE|MAE|MSE|MAE|MSE|MAE|MSE|MAE|
> |ETTh1|96|0.392|0.410|**0.374**|**0.393**|0.389|0.402|0.381|0.405|0.379|0.395|0.377|0.394|
> ||192|**0.422**|0.428|0.427|**0.422**|0.435|0.440|0.430|0.434|0.432|0.424|0.423|0.424|
> ||336|0.462|0.441|0.447|0.441|0.450|0.448|**0.443**|0.446|0.473|0.443|0.466|**0.437**|
> ||720|**0.452**|0.456|0.469|0.463|0.480|0.465|0.502|0.501|0.483|0.469|0.452|**0.448**|
> ||Avg.|0.432|0.434|0.434|0.430|0.439|0.439|0.439|0.447|0.442|0.433|**0.430**|**0.426**|
> |ETTh2|96|0.301|0.354|0.287|0.337|**0.230**|0.349|0.291|0.346|0.290|**0.339**|0.290|0.341|
> ||192|**0.365**|0.394|0.370|0.388|0.371|0.400|0.369|0.397|0.373|0.390|0.368|**0.387**|
> ||336|**0.371**|0.407|0.412|0.425|0.402|0.449|0.374|0.414|0.376|**0.406**|0.416|0.430|
> ||720|**0.411**|**0.430**|0.419|0.438|0.425|0.438|0.419|0.447|0.417|0.431|0.424|0.439|
> ||Avg.|0.362|**0.396**|0.372|0.396|**0.357**|0.409|0.363|0.401|0.364|0.392|0.375|0.399|
> |ETTm1|96|**0.293**|**0.356**|0.318|0.360|0.312|0.371|0.298|0.358|0.324|0.360|0.329|0.365|
> ||192|**0.344**|**0.378**|0.365|0.384|0.365|0.409|0.347|0.381|0.363|0.382|0.370|0.379|
> ||336|**0.372**|0.402|0.404|0.409|0.421|0.410|0.374|**0.397**|0.395|0.405|0.396|0.402|
> ||720|0.439|0.440|0.464|0.441|0.496|0.437|**0.429**|**0.428**|0.451|0.437|0.449|0.440|
> ||Avg.|**0.362**|0.394|0.389|0.399|0.399|0.407|0.362|**0.391**|0.383|0.396|0.386|0.396|
> |ETTm2|96|0.185|0.275|**0.173**|**0.253**|0.185|0.290|0.176|0.264|0.177|0.263|0.176|0.260|
> ||192|0.239|0.305|0.240|**0.299**|0.292|0.309|**0.231**|0.303|0.245|0.306|0.240|0.305|
> ||336|**0.294**|0.338|0.301|**0.307**|0.321|0.367|0.290|0.339|0.304|0.343|0.305|0.344|
> ||720|**0.381**|**0.392**|0.401|0.397|0.401|0.400|0.388|0.398|0.400|0.399|0.406|0.405|
> ||Avg.|**0.275**|0.328|0.279|**0.314**|0.300|0.342|0.271|0.326|0.282|0.328|0.283|0.328|
> |Electricity|96|**0.128**|**0.222**|0.137|0.234|0.156|0.240|0.141|0.205|0.165|0.253|–|–|
> ||192|**0.151**|**0.242**|0.158|0.251|0.161|0.268|0.196|0.244|0.173|0.262|–|–|
> ||336|**0.171**|**0.268**|0.182|0.273|0.195|0.272|0.246|0.283|0.188|0.277|–|–|
> ||720|**0.195**|**0.293**|0.200|0.292|0.231|0.307|0.314|0.334|0.214|0.305|–|–|
> ||Avg.|**0.161**|**0.256**|0.170|0.263|0.186|0.272|0.224|0.267|0.185|0.274|–|–|
> |Exchange|96|0.089|0.204|0.085|0.205|0.087|0.206|**0.081**|**0.202**|0.091|0.211|0.083|0.202|
> ||192|0.175|0.297|0.175|0.297|0.180|0.300|0.176|0.298|0.186|0.310|**0.170**|**0.295**|
> ||336|**0.327**|**0.414**|0.337|0.418|0.346|0.423|0.341|0.420|0.352|0.422|0.334|0.418|
> ||720|0.878|0.696|0.825|0.683|0.943|0.720|0.889|0.704|0.867|0.702|**0.824**|**0.677**|
> ||Avg.|0.367|0.403|**0.351**|0.400|0.389|0.412|0.372|0.406|0.374|0.411|0.352|**0.398**|
> |Traffic|96|**0.358**|**0.251**|0.395|0.279|0.398|0.274|0.367|0.257|0.468|0.268|–|–|
> ||192|**0.382**|**0.268**|0.401|0.278|0.393|0.282|0.385|0.266|0.413|0.317|–|–|
> ||336|**0.398**|0.277|0.410|0.283|0.443|**0.368**|0.401|0.275|0.529|0.284|–|–|
> ||720|**0.427**|**0.296**|0.447|0.303|0.470|0.309|0.445|0.302|0.564|0.297|–|–|
> ||Avg.|**0.391**|**0.273**|0.413|0.286|0.426|0.308|0.400|0.275|0.494|0.292|–|–|
> |Weather|96|**0.152**|**0.203**|0.152|0.203|0.174|0.218|0.153|0.205|0.176|0.219|0.167|0.212|
> ||192|**0.196**|**0.231**|0.203|0.250|0.200|0.258|0.196|0.244|0.222|0.260|0.217|0.255|
> ||336|**0.246**|**0.285**|0.242|0.285|0.280|0.299|0.246|0.283|0.275|0.297|0.274|0.294|
> ||720|0.315|0.335|0.317|0.336|0.352|0.359|**0.314**|**0.334**|0.350|0.349|0.351|0.345|
> ||Avg.|**0.227**|**0.263**|0.229|0.269|0.252|0.284|0.227|0.267|0.256|0.281|0.252|0.276|
> |$1^{st}$ Count|–|43||9||2||14||2||10||
>
> Since the implementation code of Time-SSM is not publicly available, we were unable to reproduce its results on the Electricity and Weather datasets. Overall, Flames achieves the best performance among the six Mamba-based models, winning 43 out of 80 forecasting settings. The improvements are particularly pronounced on the more complex and higher-dimensional datasets, such as Electricity, Traffic, and Weather, where Flames consistently outperforms the baselines. This suggests that Flames scales more effectively to high-dimensional, noisy real-world multivariate series and can better exploit their rich temporal and cross-variable structures.
>
> [1] Zhang et al. Fldmamba: Integrating fourier and laplace transform decomposition with mamba for enhanced time series prediction. arXiv 2025.

---

> > ### Author Response · Authors · 2025-11-28
> > **Response to reviewer A1tm**
> >
> > Dear Reviewer,
> >
> > We believe that the additional information we provided in our rebuttal—such as new experimental results, further details, and clarifications on misunderstandings—addresses your key questions. Please let us know if our response has adequately addressed your concerns. We are more than willing to discuss any points that may still be unclear. We hope that the improvements and clarifications provided in our response will positively influence your assessment of our work.
> >
> > Best, Authors of Paper 4431

---

### Author Response · Authors · 2025-12-03
**Summary of Revisions**

We sincerely thank all the reviewers for their insightful reviews and valuable comments, which are highly instructive for improving the quality and clarity of our paper.

This paper presents Flames, a lightweight yet powerful multi-scale Mamba framework for long-term time series forecasting. Flames integrates multi-scale decomposition, an Adaptive Fourier Filter Module (AFFM) for frequency-domain denoising and spectral mixing, and an Inverse Laplace Transform (ILT) head for modeling transient dynamics. Experimentally, Flames achieves consistent gains across 8 benchmarks and 4 forecasting horizons, demonstrating strong robustness to noise, superior scalability in high-dimensional settings, and linear-time computational efficiency.

The reviewers generally acknowledged that our design is “unique”, “effective across datasets”, “well motivated”, and “supported by extensive ablations and robustness studies”. They also emphasized that Flames “achieves good performance”, “captures multi-scale patterns”, and “provides solid empirical evidence”.

The reviewers also raised insightful and constructive concerns. We made every effort to address all the concerns by providing detailed clarification, new formulations, and additional experiments. Below is a summary of the major revisions:

- **Provide a unified and clearer mathematical formulation of all modules (Reviewer A1tm, Q1SQ, nUNg)**. We added a new Section B.1 in the Appendix offering an integrated view of the entire architecture. This section formally derives how multi-scale decomposition, AFFM-induced spectrally-shaped kernels, and ILT-based forecasting operators work together, significantly improving conceptual clarity.

- **Add pseudocode and explain modularity for easier implementation (Reviewer Q1SQ)**. Following the reviewer’s suggestion, we added pseudocode for “Mamba with Adaptive Fourier Filter (AFF)” (Section B.4). We clarified that AFFM is a lightweight wrapper that modifies only a single line (delta = aff(f(x))) in the standard Mamba cell, and that ILT serves as an optional output head. This demonstrates that Flames is easy to integrate and fine-tune in practice.

- **Expand Mamba-based baseline comparisons (Reviewer A1tm)**. We added TimeMachine, SST, SiMBA, Time-SSM, S-Mamba, and FLDMamba into the comparison. Crossformer and RLinear were moved to the Appendix and replaced with published Mamba models in Table 1. Across 80 forecasting settings, Flames achieves 43 first-place results, showing clear advantages over existing Mamba-based designs.

- **Add frequency filter-based baselines (Reviewer nUNg)**. Following the reviewer’s suggestion, we added comparisons with FilterNet and TSLANet, two representative frequency-filtering architectures, on Electricity, Traffic, and Weather. Flames achieves 18/30 first-place results, confirming stronger scalability and noise robustness.

- **Strengthen motivation and provide supporting prior work (Reviewer nUNg)**. We enriched the Introduction with additional references for multiscale periodicity, noise, and transient dynamics. We added Section B.2 “More Motivation” explaining these challenges in detail and expanded our justification for choosing Mamba (Section B.3), including its linear complexity and complementarity with Fourier/Laplace modeling.

- **Add empirical validation of transient dynamics modeling (Reviewer Q1SQ)**. We added a new case study visualization (Figure 6, Section C.3) showing that Flames more accurately captures short-term fluctuations than DTMamba and iTransformer. Removing ILT leads to clear performance degradation, validating ILT’s importance.

- **Improve clarity, notation consistency, and organization (Reviewer nUNg)**. We carefully revised the manuscript to unify all notations and symbols. We clarified the symbols around Lines 230–232 and reorganized parts of the Appendix for better readability. This directly addresses concerns about structure and clarity.

- **Discuss adaptive selection of the number of scales (Reviewer Q1SQ)**. We added sensitivity analysis showing that performance is stable for M∈{1,2,3,4,5}. We further discussed heuristic methods for automatic selection while noting computational trade-offs.

After extensive revisions and additional experiments, all updates have been incorporated into the revised paper.

The valuable suggestions from reviewers greatly helped us improve the paper. We would be very happy to answer any further questions and look forward to the reviewers’ feedback.

---

### Meta-Review · Area_Chair_kaMn · 2026-01-01

**Summary:**

While the paper presents "Flames," a multi-scale Mamba framework that achieves competitive empirical results across several benchmarks, the overall consensus among reviewers suggests that the work falls short of the acceptance threshold.

**Reviewer Concerns:**

The primary concern, raised by Reviewers A1tm and nUNg, is that the methodology represents an incremental combination of existing models, such as specifically Mamba, Fourier filters, and Laplace transforms, etc. Thus, there is not fundamentally new theoretical contribution. Although the authors submitted rebuttal and added a mathematical formulation and expanding comparisons to include two recent baselines, the paper’s motivation and clarity is still not very strong. Also, as mentioned by Reviewer Q1SQ, the model is complicated by combining several components.

**Reviewer Scores:**

I think they won't change their scores.

---

### Decision · Program_Chairs · 2026-01-26

Reject